# AtlasGS: Atlanta-world Guided Surface Reconstruction with Implicit Structured Gaussians

**Xiyu Zhang**[1]* **Chong Bao**[1]* **Yipeng Chen**[1] **Hongjia Zhai**[1] **Yitong Dong**[1]
**Hujun Bao**[1] **Zhaopeng Cui**[1] **Guofeng Zhang**[1]†
[1]State Key Lab of CAD & CG, Zhejiang University

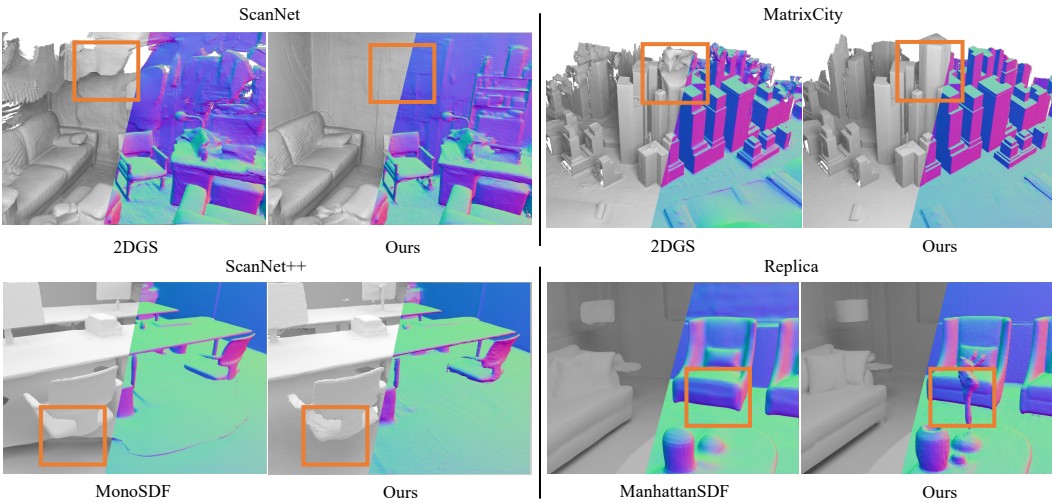

Figure 1: **Qualitative comparison of indoor and outdoor reconstruction.** For each scene, we visualize both the reconstructed geometry and the surface normals. As highlighted by the orange boxes, our method generates significantly smoother surfaces while capturing finer geometric details, clearly outperforming the compared methods.

## Abstract

3D reconstruction of indoor and urban environments is a prominent research topic with various downstream applications. However, existing geometric priors for addressing low-texture regions in indoor and urban settings often lack global consistency. Moreover, Gaussian Splatting and implicit SDF fields often suffer from discontinuities or exhibit computational inefficiencies, resulting in a loss of detail. To address these issues, we propose an Atlanta-world guided implicit-structured Gaussian Splatting that achieves smooth indoor and urban scene reconstruction while preserving high-frequency details and rendering efficiency. By leveraging the Atlanta-world model, we ensure the accurate surface reconstruction for low-texture regions, while the proposed novel implicit-structured GS representations provide smoothness without sacrificing efficiency and high-frequency details. Specifically, we propose a semantic GS representation to predict the probability of all semantic regions and deploy a structure plane regularization with learnable plane indicators for global accurate surface reconstruction. Extensive experiments demonstrate that our method outperforms state-of-the-art approaches in both indoor and urban scenes, delivering superior surface reconstruction quality.

---

*Equal contribution
†Corresponding author

39th Conference on Neural Information Processing Systems (NeurIPS 2025).

# 1 Introduction

Recently, indoor and urban 3D reconstruction from multi-view images has emerged as a popular research area, fueled by its applications in digital twins [1–3], robotic navigation [4–6], and augmented reality [7]. These technologies demand accurate and efficient reconstruction of real-world environments. However, man-made scenes often contain large low-texture planar regions, such as floors, ceilings, and unadorned walls, which pose significant challenges for image-based 3D reconstruction. Traditional multi-view stereo methods struggle on these textureless surfaces due to the lack of distinctive visual features, leading to incomplete or distorted geometry. Geometric priors play a crucial role in addressing this challenge. Monocular geometric priors provide locally smooth geometry signals for low-texture regions [8], but they often lack global consistency across different viewpoints, often resulting in inconsistent geometry like bumpy surfaces. Alternatively, the Manhattan-world assumption [9] leverages planar priors to address low-texture regions in man-made scenes but cannot be applied in urban scenes where buildings are not mutually orthogonal structures, such as the building marked by the yellow rectangle in Fig. 1.

Concurrently, 3D representation methods for indoor and urban reconstruction have evolved rapidly and achieved remarkable success. For example, 2DGS [10] employs Gaussian splitting (GS) with surfel primitives to efficiently and effectively reconstruct complex geometry. However, its discrete primitives induce discontinuities in surface reconstruction, resulting in broken surfaces in low-texture or under-observed regions, as depicted in Fig. 1. Previous implicit SDF representations [11–14] leverage the inductive continuity of coordinate-based multi-layer perceptrons (MLPs) to recover complete surfaces in these challenging regions. However, they are computationally expensive and struggle to represent complex geometries. Some methods [15–17] have explored the simultaneous learning of both representations, using implicit methods to guide GS optimization for smoother outcomes. Unfortunately, this mutual interaction often compromises reconstruction quality.

Based on these observations, we identify two critical challenges: **1)** A globally consistent geometric prior is essential to regularize low-texture regions in both indoor and urban reconstructions; **2)** A 3D representation is needed that retains the efficiency and detail-preserving capabilities of GS while incorporating the smoothness of implicit methods.

In this paper, we propose an Atlanta-world guided implicit-structured Gaussian Splatting for indoor and urban scene reconstruction.

First, the man-made indoor and urban environments can be described as an Atlanta world model [18] where there is one vertical direction aligned with gravity and multiple horizontal directions oriented from walls or urban buildings. To establish globally consistent and accurate geometry in low-texture regions, such as floor, ceiling, and walls, we incorporate this global geometric assumption into GS optimization, thereby regularizing the geometric relationships among these regions. Specifically, we propose a semantic GS representation to predict the semantic probability of floor, ceiling, and wall in the 3D scene. Besides, we design a structure plane regularization with learnable explicit plane indicators for globally accurate surface reconstruction.

Secondly, we propose a novel implicit-structured Gaussian representation, which integrates the continuity of implicit functions with the efficiency and detail preservation of 2DGS. Unlike prior works that simply overlay implicit priors on Gaussian optimization, we embed implicit voxel grids within the Gaussian Splatting framework, allowing implicit geometry to act as a smooth regularizer while maintaining high-frequency representation. Besides, we adopt a view-independent decoding on GS spatial distribution to enhance the geometric consistency across multiple viewpoints. Our representation not only improves the smoothness of surface reconstruction but also facilitates the efficient modeling of intricate geometry, striking a better balance between accuracy and efficiency.

The contribution of our paper can be summarized as: **1)** We propose a novel Atlanta-world guided implicit-structured Gaussian Splatting to achieve smooth indoor and urban scene reconstruction while preserving high-frequency details and rendering efficiency. **2)** To integrate the Atlanta-world assumption, we design a semantic GS representation to predict the semantic probability of low-texture regions such as floor, ceiling, and walls, and propose a structure plane regularization with learnable explicit plane indicators to regularize the global geometry of these low-texture regions. **3)** Extensive experiments in indoor and urban scenes demonstrate the effectiveness of our method. We show the best surface reconstruction quality quantitatively and qualitatively compared to other state-of-the-art methods.

## 2 Related Works

**Neural Implicit Surface Reconstruction.** Neural Radiance Fields [19] (NeRF) designs a neural implicit representation to reconstruct the scene from multi-view 2D images. To obtain the scene surface, some NeRF variants [11, 12, 20] combine SDF-based neural representation with volume rendering for better geometry reconstruction. In addition, to obtain better reconstruction results in some challenging scenarios, some research works introduced more prior information during the optimization process, such as geometric regularization [13], monocular depth [21–23], normal [14, 23, 24, 20], and semantics [13, 25, 26]. However, due to the limited representation capacity of MLPs [27], relying solely on MLPs may result in slow optimization and poor reconstruction performance of large scenes. Therefore, additional feature encoding is used to enhance the scene representation ability and speed up the reconstruction, such as dense feature grid [28, 22], hash table [29, 30], sparse voxel [31–33], and tetrahedron [34–38]. However, despite the use of feature encoding, all of these implicit methods still require hours of training and exhibit low inference times and insufficient detail during the reconstruction. In contrast, our method enables efficient surface reconstruction ($< 30$ minutes) with high-quality details.

**Surface Reconstruction with Gaussian Splatting.** 3D Gaussian Splatting [39] (3DGS) has emerged as a promising technique for efficient and high-quality novel view synthesis. Starting from SfM points, 3DGS represents the scene with 3D Gaussians and employs fast splatting-based rasterization to accelerate both training and inference. SuGaR [40] extends 3DGS to surface reconstruction by associating 3D Gaussians with the mesh surface and jointly optimizing both the Gaussians and the mesh. PGSR [41] utilizes planar-based Gaussian splatting combined with unbiased depth rendering to maintain global geometric accuracy. To reconstruct complete surfaces in unbounded scenes, GOF [42] leverages a ray-tracing-based volume rendering, which enables a mesh to be extracted directly from the Gaussian representation. Furthermore, 2DGS [10] and Gaussian Surfels [43] argue that the multi-view inconsistency inherent in 3DGS compromises reconstruction quality. To address this, they replace 3D Gaussians with 2D surfels, enabling more precise capture of intricate geometric details. However, the discrete nature of Gaussian Splatting undermines the smoothness of reconstructed surfaces, particularly in regions with low texture or limited observational coverage. DN-Splatter [44] employs depth and normal priors to improve the smoothness of surface reconstruction based on 3DGS. Certain methods [15, 45] simultaneously learn an implicit Signed Distance Function (SDF) field alongside 3D Gaussian Splatting, utilizing the smooth SDF field to regularize the noisy geometry inherent in 3DGS. However, the mutual interaction between these components often compromises reconstruction quality, resulting in suboptimal outcomes. In contrast, our approach integrates the implicit field with Gaussian Splatting under the Atlanta-world assumption, enabling smooth surface reconstruction while preserving high-frequency geometric details.

## 3 Methods

Our goal is to reconstruct scenes with strong structural priors from posed images. To achieve this, we first introduce the preliminaries of 2DGS [10] in Sec. 3.1, followed by our implicit-structured representation in Sec. 3.2. To leverage structural priors under the Atlanta world assumption, we propose 3D global planar regularization and 2D local surface regularization, as detailed in Sec. 3.3. Finally, we describe the training process in our framework. An overview of our approach is illustrated in Fig. 2.

### 3.1 Preliminary

Instead of representing scenes with blobs as 3D Gaussian Splatting [39], 2DGS [10] models scenes as surfels distributed around the surfaces. Each surfel is defined in a local tangent plane in world space. To render images, 2DGS rasterizes surfels to the image plane with tile-based rasterization and Ray-Splat intersection to reduce depth bias. Then, 2DGS performs alpha blending [46] to get the rendered attribute $A$ by composing these primitives sorted by depth:

$$A = \sum_i a_i \cdot \alpha_i \cdot \prod_{j=1}^{i-1}(1 - \alpha_j),\tag{1}$$

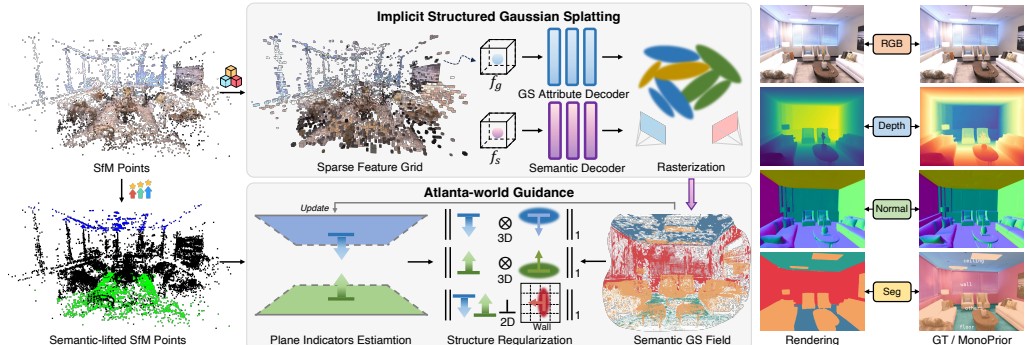

Figure 2: **Overview of AtlasGS**. Given posed images and SfM points, we construct a sparse feature grid and represent scenes as implicit-structured Gaussians. Attributes are decoded using attribute decoder and semantic decoder, followed by rasterization and supervision with RGB images, monocular geometry priors, and semantic maps. To address multiview inconsistency in textureless regions, we introduce learnable explicit plane indicators based on the Atlanta world assumption [18]. The indicators refine the global scene structure by regularizing Gaussian positions and orientations using 3D global planar and 2D local surface losses, ensuring alignment with structural elements such as walls, floors, and ceilings.

where $A$ are the rendered 2D scene information (e.g., color $c$, depth $d$, normal $\mathbf{n}$, semantics $z$), $a_i$ and $\alpha_i$ denote the 3D property and opacity contribution of $i$-th Gaussian primitive, and $\prod_{j=1}^{i-1}(1 - \alpha_j)$ is the accumulated transmittance. Then, 2DGS optimizes these Gaussians with photometric loss and distortion loss to reduce floaters. The Ray-Splat intersection and well-defined normal introduce multiview consistency depth, providing better reconstruction quality. Though 2DGS produces plausible surfaces and models high-frequency details with explicit Gaussian representation, the discrete nature introduces discontinuities, which leads to broken or protruding texture-less walls.

### 3.2 Implicit-Structured Gaussian Splatting

In this section, we present our implicit-structured Gaussian representation, which leverages a sparse feature grid and implicit functions to organize discrete Gaussian primitives, ensuring locally coherent geometry while preserving high-frequency details.

Specifically, given the sparse point clouds generated from SfM [47], we first construct a sparse feature grid $\mathcal{V} = \{\mathcal{V}_g^i, \mathcal{V}_s^i, \{\boldsymbol{\Delta}_k^i\}_{k=1}^{\mathcal{K}}, l^i\}_{i=1}^{N_v}$ with a predefined voxel size, including geometry $\mathcal{V}_g^i$ and semantic features $\mathcal{V}_s^i$, offsets of $\mathcal{K}$ local Gaussians $\{\boldsymbol{\Delta}_k^i\}_{k=1}^{\mathcal{K}}$, scaling factor $\{l^i\}_{i=1}^{N_v}$ shared with local Gaussians. Given a voxel located in $\mathbf{v}$, we deocde all Gaussian geometry attributes of $\mathcal{K}$ local Gaussians via corresponding geometry MLP $\mathcal{M}_g(\cdot)$ and semantic MLP $\mathcal{M}_s(\cdot)$

$$\{\alpha_k, s_k, q_k\}_{\mathcal{K}}, \{z_k\}_{\mathcal{K}} = \mathcal{M}_g(\mathcal{V}_g(\mathbf{v})), \mathcal{M}_s(\mathcal{V}_s(\mathbf{v}))$$
$$\{c_k\}_{\mathcal{K}} = \mathcal{M}_g(\mathcal{V}_g(\mathbf{v}), \mathbf{d}). \tag{2}$$

Here, $\mathbf{d}$ is view direction which assists in capturing view-dependent appearance, $\alpha \in \mathbb{R}$, $s \in \mathbb{R}^2$, $q \in \mathbb{R}^4$, $c \in \mathbb{R}^3$, and $z \in \mathbb{R}^4$ refer to the opacity, scale, rotation, color, and semantic attributes of each Gaussian, respectively, whose positions $\mathbf{p}$ is defined as $\mathbf{p}_k^i = \mathbf{v}_i + l \cdot \boldsymbol{\Delta}_k^i$. Then we render images with all these Gaussian attributes decoded from the sparse feature grid by surfel rasterization from [10]. In contrast to 2DGS [10], which optimizes Gaussians independently, our decoder predicts local Gaussian attributes, allowing each Gaussian's optimization to influence its neighbors and capture fine object details through the predicted Gaussian primitives. Consequently, the implicit-structured Gaussian framework combines the strengths of MLPs and Gaussians, achieving smooth local geometry while preserving high-frequency details.

**Gaussian Semantic Lifting.** To recognize structural regions in scenes, we lift 2D semantics to each Gaussian. We use the 2D pseudo-labels $\hat{Z}$ obtained by the pre-trained semantic segmentation model [48] as supervision and optimize the semantic Gaussians with the rendered semantic probability $Z$. To obtain the rendered semantics probability $Z$, we render the 3D semantic attribute $z$ defined

in Eq. (2) into image space with Eq. (1), and optimize the semantic feature grid $\mathcal{V}_s$ and semantic MLP $\mathcal{M}_s$ by minimizing the cross-entropy loss:

$$Z = \sum_i z_i \cdot \text{sg}[\alpha_i \cdot \prod_{j=1}^{i-1}(1 - \alpha_j)], \tag{3}$$

$$\mathcal{L}_{\text{sem}} = -\frac{1}{|\mathcal{U}|} \sum_{u \in \mathcal{U}} \hat{Z}(u) \cdot \log Z(u), \tag{4}$$

where $\text{sg}[\cdot]$ denotes the stop-gradient operator, used to prevent inconsistent supervision from distorting the geometry, and $\mathcal{U}$ denotes the set of training pixels in each iteration. Here, we define $z \in \mathbb{R}^4$, indicating wall, floor, ceiling, and others.

### 3.3 Atlanta World Guided Planar Regularization

Man-made indoor and urban environments typically exhibit rich structural information and conform to the Atlanta world assumption [18]. To leverage the globally consistent structural priors inherent to such scenes, we first propose learnable explicit plane indicators to effectively represent scene structural information, such as the ceiling and floor. We then introduce two types of regularization: 3D global planar regularization, which refines Gaussian positions and orientations to align with the plane indicators, and 2D local surface regularization, which provides positional supervision in poorly defined wall regions by aligning them with the plane indicators.

**Explicit Plane Indicators**  The Atlanta world assumption [18] hypothesizes that such a scene can be approximated by a combination of a dominant horizontal plane, like the floor for indoor scenes and ground for outdoor scenes, and multiple vertical planes, such as walls and buildings. Based on this assumption, we define explicit plane indicators with a gravity direction and two distance offsets to represent the floor and ceiling of an indoor scene. Specifically, the floor plane and ceiling plane can be represented with $\pi_f = (\mathbf{n}_g, d_f), \pi_c = (-\mathbf{n}_g, d_c)$, and their plane parametric equations are as follows:

$$d_f + \mathbf{n}_g \cdot \mathbf{p} = \mathbf{0}, \quad d_c - \mathbf{n}_g \cdot \mathbf{p} = \mathbf{0}, \tag{5}$$

where $\mathbf{p} \in \mathbb{R}^3$ is the location of 3D points, $d_f$ and $d_c$ are the distances from the origin to their respective planes, and $\mathbf{n}_g$ is the gravity direction. For the urban scenes, we omit the ceiling plane.

To achieve this, we extract ceiling points and floor points, either from semantic Gaussians or semantic lifted sparse points. We then apply RANSAC [49] to regress the plane coefficients and determine the gravity direction to initialize plane indicators. Finally, we optimize the plane indicators alongside the Gaussians. Please refer to our supplementary for further details.

**3D Global Planar Regularization.**  According to the associated probability distribution $z \in \mathbb{R}^4$, we can obtain the semantic label of each Gaussian. We enforce two critical geometric regularizations: normal alignment and planar constraint. The normal alignment enforces normals to be perpendicular to the gravity direction in wall regions while ensuring they remain parallel in ceiling and floor regions. Besides, the planar constraint ensures that the Gaussian positions $\mathbf{p}_i$ lie in their corresponding ceiling or ground plane. Thus, the 3D global planar regularization is formulated as:

$$\mathcal{L}_{\text{3D}} = \sum_{i \in M_{\parallel}} p_{\parallel}(1 - |\mathbf{n}_g^{\top}\mathbf{n}_i|)$$
$$+ \sum_{i \in M_{\perp}} p_{\perp}|\mathbf{n}_g^{\top}\mathbf{n}_i| + \sum_{i \in M_c} p_c|d_c - \mathbf{n}_g^{\top}\mathbf{p}_i| + \sum_{i \in M_f} p_f|d_f + \mathbf{n}_g^{\top}\mathbf{p}_i|, \tag{6}$$

where $M_{\text{f}}, M_{\text{c}}, M_{\parallel}$ and $M_{\perp}$ are the floor, ceiling, parallel, perpendicular sets, $p_{\parallel}, p_{\perp}, p_c, p_f$ are the corresponding probabilities, $\mathbf{n}_i$ and $\mathbf{p}_i$ are the normal and position of $i$-th Gaussian.

**2D Local Surface Regularization.**  Previous methods [14, 11, 13] with implicit representation derive normals from the gradient of the given points, while optimizing normals can also optimize the

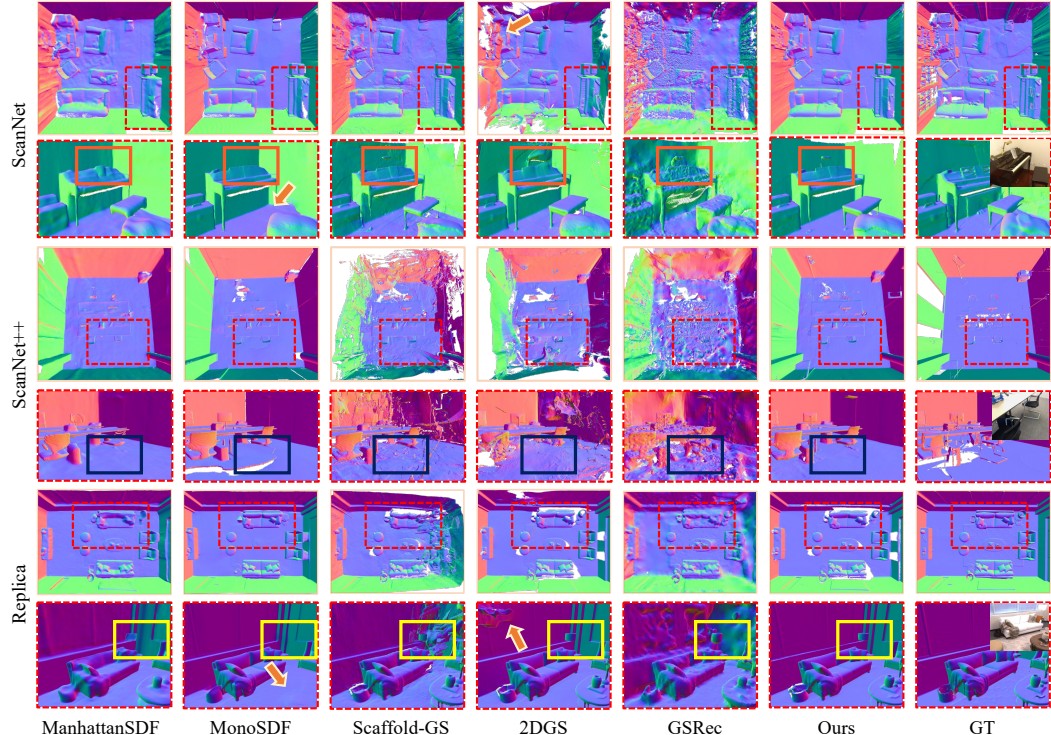

Figure 3: **Qualitative comparison of indoor scene reconstruction**. We show the reconstruction performance of the baselines and our approach on ScanNet [50], ScanNet++ [51], and Replica [52] datasets. As highlighted in the boxes, our approach maintains local smoothness and preserves high frequency. The red dashed boxes mark regions that are zoomed in below for closer inspection.

local surface. However, the Gaussian representation explicitly decouples the positions and normals of Gaussians. It poses a challenge that only optimizing Gaussian orientations in 3D space does not directly affect their spatial distribution. Without explicit positional regularization, wall Gaussians may not lie on the same plane, resulting in misalignment between the surface and the plane indicators.

To address this issue, we introduce 2D local surface regularization by regularizing the normal $\mathbf{N}_d$ from the rendered depth $D$. With the semantic Gaussians, we obtain coherently rendered semantics and depth, from which we derive local surface normals in wall regions. Then, we align the surface normal with our plane indicators, optimizing Gaussian positions more directly. Besides, to mitigate misclassification introduced by the semantic segmentation model [48], we weigh the loss according to the probabilities. Thus, our 2D local surface regularization loss is formulated as:

$$\mathcal{L}_{2D} = \sum_{i \in M_\perp} p_w \left( |\mathbf{N}_d \cdot \mathbf{n}_g| \right) + p_{c,f} \left( 1 - |\mathbf{N}_d \cdot \mathbf{n}_g| \right), \tag{7}$$

$$\mathbf{N}_d = \frac{\nabla_x \mathbf{P} \times \nabla_y \mathbf{P}}{|\nabla_x \mathbf{P} \times \nabla_y \mathbf{P}|}, \tag{8}$$

where $p_{c,f}$ is the sum of the probabilities of the floor and ceiling, $p_w$ is the probability of the wall, $\mathbf{P}$ are the points backprojected from depth, $\nabla$ is the gradient operator. So, our structural regularization loss is:

$$\mathcal{L}_{\text{reg}} = \mathcal{L}_{3D} + \mathcal{L}_{2D}. \tag{9}$$

### 3.4 Training

We optimize Gaussians and structure with the following:

$$\mathcal{L} = \mathcal{L}_{\text{rgb}} + \lambda_1 \mathcal{L}_{\text{depth}} + \lambda_2 \mathcal{L}_{\text{normal}} + \lambda_3 \mathcal{L}_{\text{reg}} + \lambda_4 \mathcal{L}_{\text{sem}} + \lambda_5 \mathcal{L}_{\text{dist}} + \lambda_6 \mathcal{L}_{\text{nc}}, \tag{10}$$

where $\mathcal{L}_{\text{rgb}}$ is photometric loss proposed in original 3DGS, the distortion loss $\mathcal{L}_{\text{dist}}$ and normal consistency loss $\mathcal{L}_{\text{nc}}$ from 2DGS [10]. $\lambda_i, i \in \{1 \dots 6\}$ are the loss weight, $\mathcal{L}_{\text{sem}}$ is the semantic

Table 1: **Quantitative comparison on Replica[52] and ScanNet++ [51]**. We report accuracy (Acc) and completeness (Comp) in cm, others in percentage with a 5cm threshold. The first three results are highlighted in red, orange, and yellow, respectively.

| Category | Methods | Replica [52] | | | | | ScanNet++ [51] | | | | |
|---|---|---|---|---|---|---|---|---|---|---|---|
| | | Acc↓ | Comp↓ | Prec↑ | Recall↑ | F-score↑ | Acc↓ | Comp↓ | Prec↑ | Recall↑ | F-score↑ |
| Implicit | ManhattanSDF [13] | 4.76 | 5.59 | 68.80 | 66.40 | 67.57 | 3.96 | 4.98 | 77.30 | 76.16 | 76.67 |
| | MonoSDF [14] | 4.14 | 5.38 | 75.50 | 70.89 | 73.08 | 4.05 | 5.57 | 76.25 | 75.16 | 75.65 |
| Explicit | Scaffold-GS [54] | 8.58 | 11.27 | 63.53 | 54.91 | 58.89 | 23.11 | 15.36 | 28.18 | 31.75 | 29.78 |
| | 2DGS [10] | 4.76 | 6.34 | 74.54 | 65.37 | 69.64 | 16.53 | 17.91 | 22.84 | 20.79 | 21.71 |
| | DN-Splatter [44] | 16.97 | 15.52 | 32.67 | 30.90 | 31.75 | 16.77 | 15.05 | 22.42 | 21.97 | 22.16 |
| | GSRec [17] | 4.90 | 6.89 | 73.32 | 67.69 | 70.37 | 9.37 | 9.13 | 47.12 | 53.81 | 50.14 |
| | Ours | 2.25 | 4.08 | 93.18 | 82.22 | 87.35 | 3.22 | 4.09 | 87.59 | 87.47 | 87.48 |

Table 2: **Quantitative comparison on ScanNet [50]**. We report accuracy (Acc) and completeness (Comp) in cm, others in percentage with a 5cm threshold. The first three results are highlighted in red, orange, and yellow, respectively.

| Category | Methods | Acc↓ | Comp↓ | Prec↑ | Recall↑ | F-score↑ | Time ↓ | FPS ↑ |
|---|---|---|---|---|---|---|---|---|
| Implicit | ManhattanSDF [13] | 4.25 | 5.23 | 72.39 | 63.18 | 67.25 | > 7 h | < 10 |
| | MonoSDF [14] | 4.25 | 4.76 | 73.53 | 69.18 | 71.21 | > 7 h | < 10 |
| Explicit | Scaffold-GS [54] | 9.47 | 7.99 | 51.41 | 49.08 | 50.17 | 12 mins | 279 |
| | 2DGS [10] | 11.46 | 13.89 | 43.15 | 36.17 | 39.27 | 11 mins | 118 |
| | DN-Splatter [44] | 13.54 | 14.77 | 21.71 | 18.94 | 20.22 | 12 mins | 145 |
| | GSRec [17] | 6.71 | 5.40 | 60.36 | 66.63 | 63.30 | 35 mins | 261 |
| | Ours | 3.62 | 3.93 | 80.31 | 75.85 | 77.98 | 27 mins | 70 |

loss mentioned in Sec. 3.2. To provide the smoothness of local surfaces, we incorporate monocular geometry priors from pre-trained models [8, 53] for indoor reconstruction during training.

$$\mathcal{L}_{\text{depth}} = \sum ||(wD + q) - \hat{D}||^2, \tag{11}$$

$$\mathcal{L}_{\text{normal}} = \sum \left(1 - \mathbf{N} \cdot \hat{\mathbf{N}}\right) + \left(1 - \mathbf{N}_d \cdot \hat{\mathbf{N}}\right), \tag{12}$$

where $\mathbf{N}, \hat{\mathbf{N}}, \hat{D}$ are the rendered normal, the normal prior and depth prior from [8, 53], respectively. $w, q$ are the scale and shift computed from least-squares to align rendered depth $D$ to monocular depth $\hat{D}$.

## 4 Experiments

**Implementation Details.** We implement our approach in PyTorch [55], incorporating a custom surfel rasterization module for semantic learning, and optimize the parameters with Adam optimizer [56]. The hyperparameter $\mathcal{K}$ is set to 10, and the voxel size is fixed at 0.01. During training, the loss weights $\lambda_1, \lambda_2, \lambda_3, \lambda_4, \lambda_5, \lambda_6$ are set to 0.25, 0.1, 0.1, 1.0, 100, and 0.05, respectively. The explicit plane indicator is initially derived from the semantic lifted SfM points and is reinitialized using the semantic Gaussians if the discrepancy exceeds a predefined threshold. Surfaces are extracted using TSDF Fusion [57]. All experiments are performed on a single NVIDIA 4090D GPU. Additional implementation details are provided in the supplementary material.

**Datasets.** Our method leverages the Atlanta world assumption, making it well-suited for both indoor and urban scenes. We evaluate our method using well-known datasets for both indoor and outdoor scene reconstruction. For indoor environments, we use ScanNet [50], ScanNet++ [51], and Replica [52]. For outdoor settings, we employ the MatrixCity [58] dataset for surface reconstruction. ScanNet and ScanNet++ feature large-scale RGB-D images and 3D surfaces akin to real-world scenarios, while Replica offers a synthetic dataset with high-quality images from 3D meshes. MatrixCity provides a synthetic, city-scale dataset for neural rendering. In line with previous studies [13, 14], we select four scenes from ScanNet, seven from Replica, and four from ScanNet++, sampling images

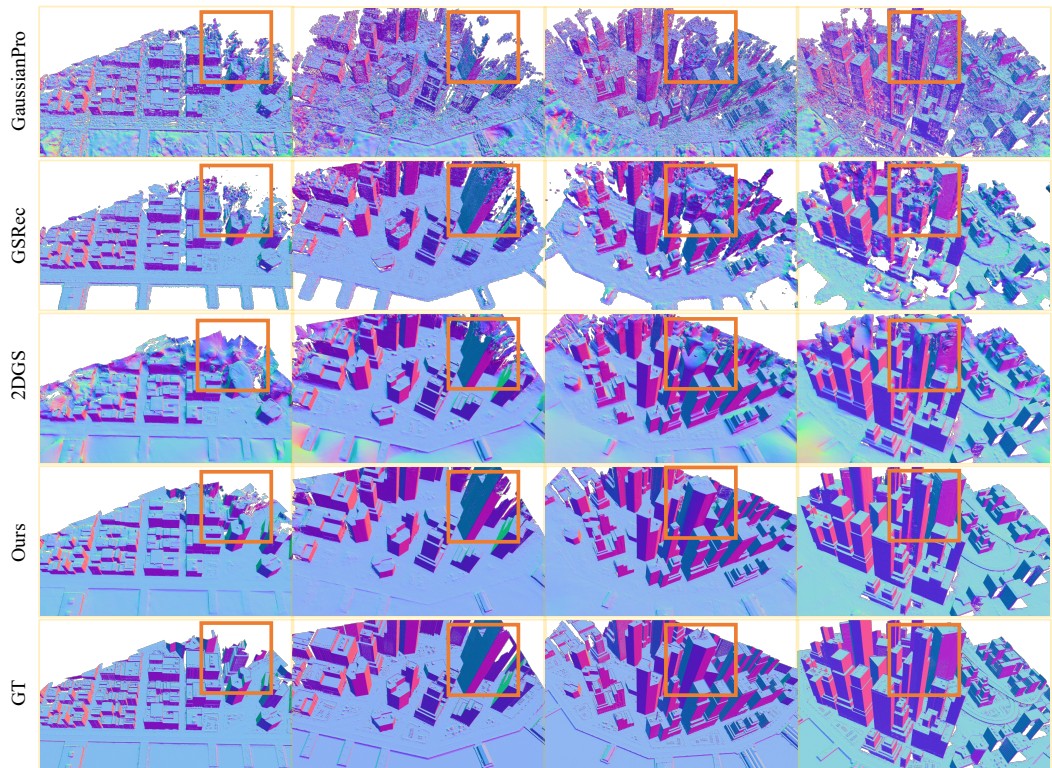

Figure 4: **Qualitative comparison of outdoor scene reconstruction**. As highlighted in the boxes, our approach can produce detailed and noise-free surfaces in textureless regions.

uniformly from the sequences. For outdoor evaluation, four city blocks from MatrixCity are used. Additional details are provided in the supplementary material.

**Baselines.** For indoor scenes, we compare our approach against two types of methods: 1) Neural implicit representations: ManhattanSDF [13] and MonoSDF [14]; 2) Gaussian-based representations: Scaffold-GS [54], 2DGS [10], DN-Splatter [44], and GSRec [17]. Additional supervision with monocular depth and normals, as noted in Eq (11) and Eq (12), is integrated into Scaffold-GS and 2DGS for indoor scenes. The quantitative evaluation for indoor scenes includes accuracy, completeness, precision, recall, and F-score. For outdoor scenes, comparisons are made with 2DGS, GSRec, Scaffold-GS, and GaussianPro [59]. In this context, the geometric evaluation relies on accuracy, completeness, and chamfer distance.

## 4.1 Comparison

**Indoor Surface Reconstruction.** We evaluate surface reconstruction on ScanNet [50], Scan-Net++ [51], and Replica [52]. As shown in Tab. 1 and 2, our method outperforms both implicit and explicit baselines, achieving state-of-the-art results. SDF-based implicit methods like Manhattan-SDF [13] and MonoSDF [14] ensure smooth surfaces but struggle with fine details, e.g., "lamp" in orange box of Fig. 3) and suffer from multiview inconsistency in textureless regions, e.g., there is a discontinuity on the floor in MonoSDF.

Table 3: **Quantitative comparison on Matrix-City [58].**

| Methods | Acc↓ | Comp↓ | CD↓ |
|---|---|---|---|
| GaussianPro [59] | 0.102 | 0.081 | 0.091 |
| Scaffold-GS [54] | 0.328 | 0.303 | 0.316 |
| GSRec [17] | 0.048 | 0.175 | 0.112 |
| 2DGS [10] | 0.115 | 0.098 | 0.106 |
| Ours | 0.022 | 0.034 | 0.028 |

Additionally, the time-consuming ray sampling in NeRF-based frameworks is to blame for the training time exceeding 7 hours and the inability to achieve real-time rendering. Gaussian-based explicit methods, such as Scaffold-GS and 2DGS, are fast but produce discontinuous surfaces due to

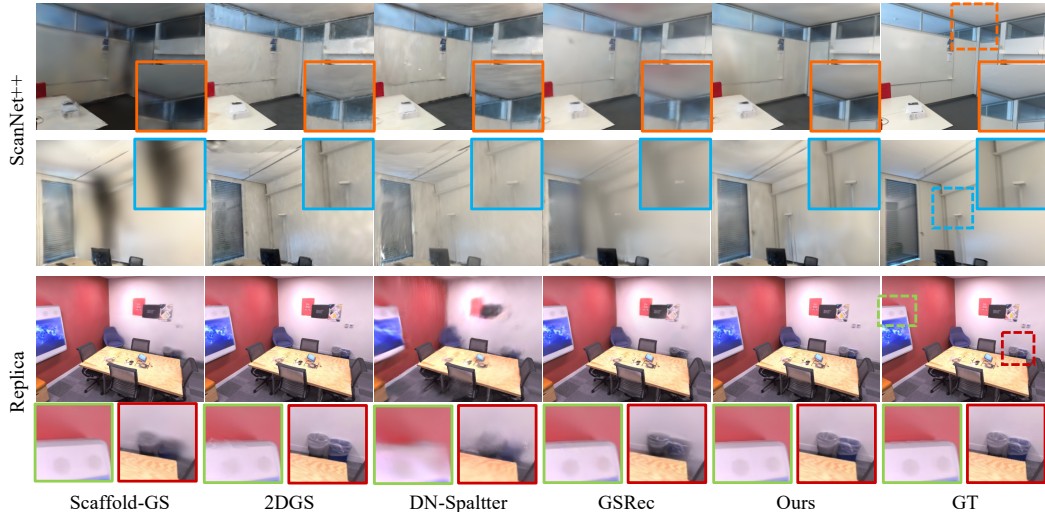

Figure 5: **Qualitative comparison of novel view synthesis**. We show the novel view synthesis results of different Gaussian splatting-based approaches on ScanNet++ [51] and Replica [52] datasets. Our method can obtain higher-fidelity rendering results with less noisy information than the baselines.

Table 4: **Quantitative comparison of novel view synthesis**. We perform experiments on Replica [52] and ScanNet++ [51] datasets.

| Methods | Replica [52] | | | ScanNet++ [51] | | |
|---|---|---|---|---|---|---|
| | PSNR↑ | SSIM↑ | LPIPS↓ | PSNR↑ | SSIM↑ | LPIPS↓ |
| ScaffoldGS [54] | 38.08 | 0.9660 | 0.0961 | 18.25 | 0.7749 | 0.2764 |
| 2DGS [10] | 41.59 | 0.9823 | 0.0464 | 21.87 | 0.8114 | 0.3060 |
| DN-Splatter [44] | 29.02 | 0.8967 | 0.2312 | 22.76 | 0.8226 | 0.2971 |
| GSRec [17] | 36.00 | 0.9574 | 0.1205 | 22.96 | 0.8314 | 0.2708 |
| Ours | 39.58 | 0.9756 | 0.0766 | 22.51 | 0.8321 | 0.2517 |

independent primitive optimization or view-dependent geometry. GSRec [17] improves geometry via IMLS but still yields noisy results. In contrast, our implicit-structured Gaussians combine locally coherent geometry with high-frequency detail preservation, enabling smoother and more accurate reconstructions. While slower than prior Gaussian methods due to decoding all Gaussians when rendering, our approach remains much faster than implicit ones and delivers superior quality.

**Urban Surface Reconstruction.**    Structural priors are common in man-made environments, including both indoor scenes and urban buildings. To evaluate our method under such priors, we use the MatrixCity dataset [58]. We compare against GaussianPro [59], Scaffold-GS [54], GSRec [17], and 2DGS [10]. As shown in Tab. 3, our method yields more accurate and complete surfaces by leveraging the Atlanta world assumption. In Fig. 4, GaussianPro suffers from noisy surfaces and inconsistent depth despite normal propagation. GSRec reduces noise via IMLS regularization but produces sparse reconstructions in textureless regions, and its Poisson surface reconstruction trims low-density areas, resulting in missing geometry. 2DGS also struggles in textureless regions like the sea and building facades, leading to artifacts such as protrusions or holes. In contrast, our approach delivers smoother and more accurate surfaces in these challenging regions.

**Novel View Synthesis.**    We evaluate novel view synthesis on the Replica [52] and ScanNet++ [51] datasets. As demonstrated in Tab. 4 and Fig. 5, our method achieves superior quantitative results, rendering photorealistic views with accurate geometry while avoiding the artifacts common in other approaches. While Scaffold-GS and 2DGS perform well on synthetic datasets without significant lighting variations, they struggle to render photorealistic novel views in real scenes. Scaffold-GS models significant lighting variations using view-dependent geometry, which can lead to overfitting in scenes with substantial lighting changes, such as those in [51]. This overfitting results in inaccurate lighting environment and geometry, as shown in Fig. 5. 2DGS [10] achieves a higher quantitative result on Replica with a discrete representation, however, the discrete representation exhibits protrud-

ing surfaces and results in noisy images on real scenes. GSRec [17] improves geometric accuracy but produces a blurry background and objects, lacking detailed modeling. In contrast, with its precise geometry, our method effectively models lighting variations across views while accurately capturing the appearance of the background and objects.

## 4.2 Ablation Study

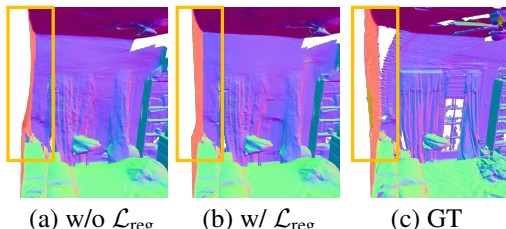

(a) w/o $\mathcal{L}_{\text{reg}}$     (b) w/ $\mathcal{L}_{\text{reg}}$     (c) GT

**Figure 6: Qualitative ablation on ScanNet [50].** (a) Ours w/o $\mathcal{L}_{\text{reg}}$ (Row b in Tab. 5); (b) Ours w/ $\mathcal{L}_{\text{reg}}$ (Row f in Tab. 5); (c) GT.

Table 5: **Quantitative ablation** on ScanNet [50].

| Merthod | $\mathcal{L}_{\text{depth}}$ | $\mathcal{L}_{\text{normal}}$ | $\mathcal{L}_{\text{3D}}$ | $\mathcal{L}_{\text{2D}}$ | CD↓ | F-score↑ |
|---|---|---|---|---|---|---|
| a) 2DGS | ✓ | ✓ | ✗ | ✗ | 12.68 | 39.27 |
| b) Ours | ✓ | ✓ | ✓ | ✓ | 3.77 | 77.98 |
| c) w/o $\mathcal{L}_{\text{2D}}$ | ✓ | ✓ | ✓ | ✗ | 3.97 | 75.52 |
| d) w/o $\mathcal{L}_{\text{reg}}$ | ✓ | ✓ | ✗ | ✗ | 4.10 | 74.23 |
| e) w/o $\mathcal{L}_{\text{normal}}$ | ✓ | ✗ | ✓ | ✓ | 3.89 | 76.30 |
| f) w/o $\mathcal{L}_{\text{depth}}$ | ✗ | ✓ | ✓ | ✓ | 4.23 | 74.22 |

We conduct an ablation study on the ScanNet [50] dataset to evaluate the contribution of each component in our method, including the implicit-structured Gaussian, depth and normal priors, as well as the proposed 3D global planar and 2D local surface regularization terms. We report geometric evaluation metrics including Chamfer Distance (CD) and F-score (5 cm threshold), as summarized in Tab. 5.

As shown in Tab. 5, each component contributes positively to surface reconstruction quality. Compared to 2DGS (Row a) which uses normal and depth priors, our method with implicit-structured Gaussians (Row d) significantly improves geometry, as it provides locally coherent structures and better leverages the guidance from geometry priors. By comparing Rows b, c, and d, we observe that our 3D global planar regularization and 2D local surface regularization enhance geometry quality and yield superior geometric metrics. These regularization terms ensure globally consistent geometric supervision under the Atlanta world assumption, mitigating the inconsistencies and inaccuracies inherent in depth and normal priors. Furthermore, Fig. 6 illustrates the geometric comparison of our structural regularization, demonstrating straighter wall regions. Additionally, we perform ablations on the geometry priors themselves. As seen in rows e and f, removing either the normal prior or the depth prior leads to noticeable degradation, confirming that both types of geometric supervision are critical for high-quality reconstruction. Our full model (Row b) integrates all components and achieves the best performance with the lowest CD and highest F-score.

## 5 Conclusion & Limitations

This work presents a novel framework, **AtlasGS**, which reconstructs structured scenes using implicit-structured Gaussians under the Atlanta world assumption. We introduce our implicit-structured Gaussian representation as a hybrid approach that provides locally coherent geometry via MLPs and preserves high-frequency details through explicit Gaussian representation. To resolve the global inconsistency of geometric priors, we propose 3D and 2D regularization strategies based on the Atlanta world assumption, effectively correcting textureless regions. As a result, our method achieves fast and accurate surface reconstruction, and extensive experiments demonstrate that it achieves state-of-the-art performance. However, there are also some limitations in our work. First, the training and rendering speed are slower than the previous Gaussian-based methods [54, 10, 39]. Second, our method is primarily based on the Atlanta world assumptions, which depend on a pretrained semantic segmentation model for a limited set of elements. The future extension of our work is to speed up the training and rendering speed and facilitate SAM [60] and geometry priors to provide planar information, which can provide better performance and broader applicability.

## Acknowledgement

This work was partially supported by NSF of China (No. 62425209).

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

# A Appendix

## A.1 Planar Indicator Initialization

As described in Sec. 3.3, we initialize the plane indicator using ceiling and floor points derived from semantically lifted SfM points or semantic Gaussian primitives. The SfM points are generated by triangulating posed images through 2D feature matching, establishing 2D-3D correspondences. Utilizing these correspondences, we aggregate semantic labels for each 3D point from 2D semantic maps across multiviews and apply a voting procedure to identify the most prevalent semantic label, including those for ceiling and floor. The Gaussian semantic lifting module, mentioned in Sec. 3.2, lifts 2D semantic maps to each Gaussian primitive, and each primitive contains a semantic probability of the wall, floor, ceiling, or other categories. Consequently, SfM points and Gaussian primitives are assigned structural semantic labels such as wall, floor, or ceiling, allowing us to extract ceiling and floor points.

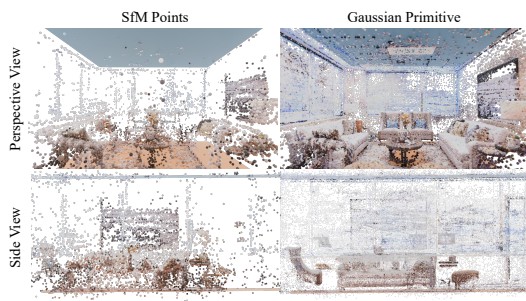

Figure 7: **Plane Indicator Visualization**. We visualize the plane indicators derived from both the semantic lifted SfM points and the semantic Gaussian primitives from both the perspective and side views. In the visualization, the ceiling plane is colored in blue, while the floor plane is colored in orange.

Subsequently, we conduct plane fitting to identify the floor plane $(\mathbf{n}_f, d_f)$ using RANSAC [49] applied to the extracted floor points. The normal vector $\mathbf{n}_f$ is chosen as the gravity direction $\mathbf{n}_g$. The offset of the ceiling plane, $d_c$, is calculated based on the ceiling points and the gravity direction as follows:

$$d_c = \operatorname*{mean}_{\mathbf{p} \in \mathbf{P}_{\text{ceiling}}} (\mathbf{n}_g \cdot \mathbf{p}), \tag{13}$$

where $\mathbf{P}_{\text{ceiling}}$ represents the set of ceiling points. The plane indicator is initially determined using the semantic lifted points. If the angle deviation or the offset discrepancy surpasses a threshold, the plane indicator is reinitialized using semantic Gaussian primitives to minimize inaccuracies in textureless regions. Fig. 7 further illustrates plane indicators derived from both semantic lifted sparse points and semantic Gaussian primitives, demonstrating that both approaches can provide reliable structural priors.

## A.2 Additional Implementation Details

Our implementation is based on PyTorch, utilizing customized surfel rasterization techniques for semantic learning. Parameters are optimized using the Adam optimizer. Most of the training learning rates are similar to those used in [54]. We set the hyperparameter $\mathcal{K}$ to 10 for indoor scenes and 5 for urban scenes, with a voxel size of 0.01, and the feature dim is 32 in our sparse feature grid. For all scenes, the implicit-structured Gaussian is trained for 40,000 steps. Voxels grow between steps 1,500 and 20,000, provided the gradients of the Gaussians exceed 2e-4 and are pruned if the opacities of all local Gaussians fall below 0.005. During training, we start our 3D global planar regularization from step 7000 and 2D local surface regularization from 20000. After completing training, surfaces are extracted using TSDF-Fusion [57], following the approach described in [10].

Table 6: **Defination of metrics**. $P$ and $P^*$ are the 3D points from the predicted and the GT mesh.

| Metric | Definition |
|---|---|
| Acc | $\operatorname{mean}_{\boldsymbol{p} \in P} (\min_{\boldsymbol{p}^* \in P^*} \|\boldsymbol{p} - \boldsymbol{p}^*\|)$ |
| Comp | $\operatorname{mean}_{\boldsymbol{p}^* \in P^*} (\min_{\boldsymbol{p} \in P} \|\boldsymbol{p} - \boldsymbol{p}^*\|)$ |
| CD | $\frac{\text{Acc} + \text{Comp}}{2}$ |
| Prec | $\operatorname{mean}_{\boldsymbol{p} \in P} (\min_{\boldsymbol{p}^* \in P^*} \|\boldsymbol{p} - \boldsymbol{p}^*\| < 0.05)$ |
| Recall | $\operatorname{mean}_{\boldsymbol{p}^* \in P^*} (\min_{\boldsymbol{p} \in P} \|\boldsymbol{p} - \boldsymbol{p}^*\| < 0.05)$ |
| F1-score | $\frac{2 \times \text{Prec} \times \text{Recall}}{\text{Prec} + \text{Recall}}$ |

## A.3 Additional Exprimental Details

Similar to previous works for indoor scene reconstruction [14], we select four scenes in ScanNet [50], including *scene0050_00, scene0084_00, scene0580_00, scene0616_00* and seven scenes

in Replica [52], *office0~office3*, *room0~room2*, and as for ScanNet++ [51], we select four scenes, *8b5caf3398, b20a261fdf, f34d532901, f6659a3107*.

As described in Sec. 4, we uniformly sample images on the indoor scenes due to redundant images in the original dataset. For each scene in ScanNet [50] and Replica [52], we select one out of every 10 images in the original image sequence. For ScanNet++ [51], we use the image sequence from the iPhone and select one out of every 60 images. All the images are cropped and resized, and center-cropped to 640×480. For MatrixCity [58], we use all the provided images and make the image resolution 960×540. The SfM points are triangulated by COLMAP [47] with given images and corresponding poses.

## A.4 Evaluation Metrics

Following previous methods [13, 14], we evaluate accuracy (Acc), completeness (Comp), Chamfer Distance (CD), precision (Prec), recall (Recall), and F1-score on ScanNet [50], ScanNet++ [51], and Replica [52]. Tab. 6 shows the definition of these metrics.

## A.5 Additional Indoor Experiments

**Semantic Segmentation.** We evaluate the semantics from the rendered and the pre-trained segmentation model Mask2Former [48] on Replica [52] and ScanNet++ [51]. As shown in Tab. 7, ours achieves better results across all three classes on both datasets. By leveraging Gaussian semantic lifting, our model effectively aggregates multi-view information into 3D space and renders view-consistent semantic maps. In contrast, the 2D semantic segmentation model is more susceptible to image noise, leading to misclassifications, as illustrated in Fig. 8. The joint optimization scheme also helps correct semantic misclassifications, particularly around the boundaries between floors and walls.

Table 7: **Quantitative comparison of structural layout segmentation on Replica [52] and ScanNet++ [51] dataset.**

| Methods | Replica [52] | | | ScanNet++ [51] | | |
|---|---|---|---|---|---|---|
| | $IoU_w$ ↑ | $IoU_f$ ↑ | $IoU_c$ ↑ | $IoU_w$ ↑ | $IoU_f$ ↑ | $IoU_c$ ↑ |
| Mask2Former [48] | 0.628 | 0.823 | 0.900 | 0.684 | 0.780 | 0.767 |
| Ours | **0.701** | **0.846** | **0.927** | **0.732** | **0.858** | **0.777** |

GT          Mask2Former          Ours

Figure 8: **Qualitative comparison of structural layout segmentation**.

## A.6 Additional Qualitative Results

We present qualitative top-view results for ScanNet, ScanNet++, and Replica in Figs. 9 to 11, respectively. For additional comparisons, please refer to our accompanying video.

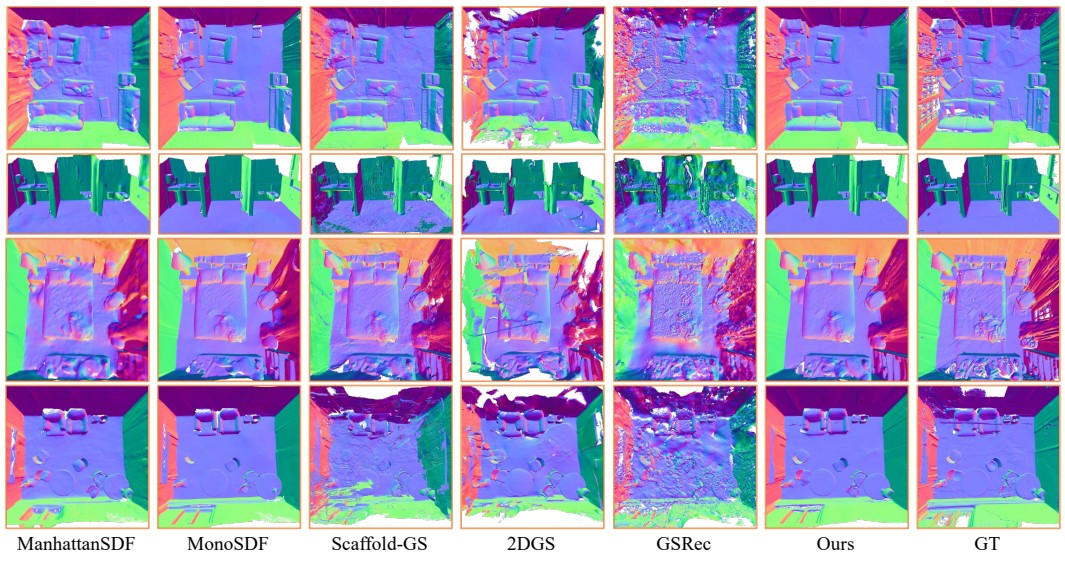

| ManhattanSDF | MonoSDF | Scaffold-GS | 2DGS | GSRec | Ours | GT |

Figure 9: **Qualitative comparison of surface reconstruction on ScanNet [50].**

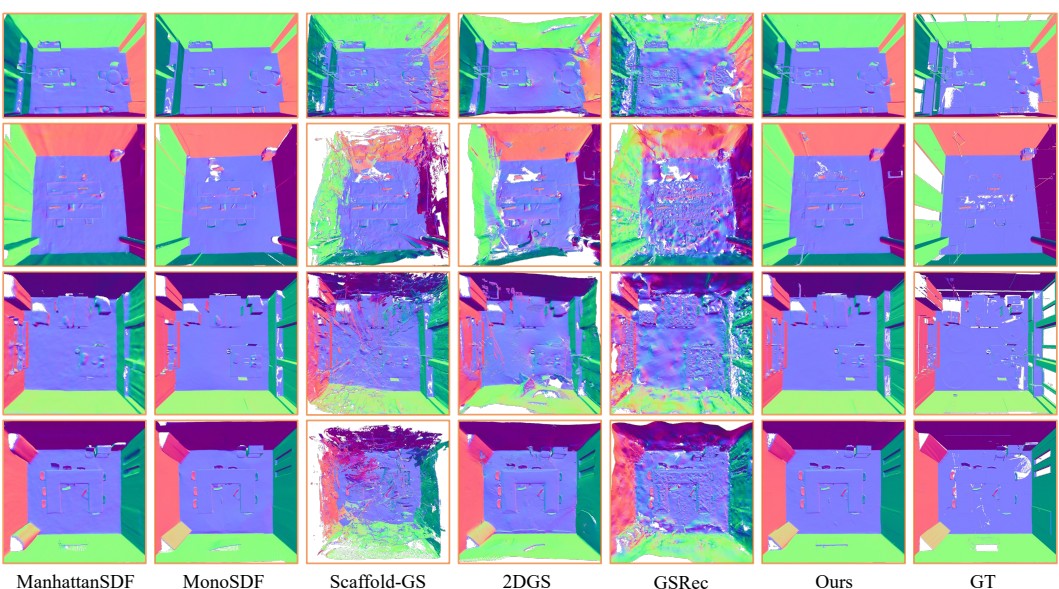

| ManhattanSDF | MonoSDF | Scaffold-GS | 2DGS | GSRec | Ours | GT |

Figure 10: **Qualitative comparison of surface reconstruction on ScanNet++ [51].**

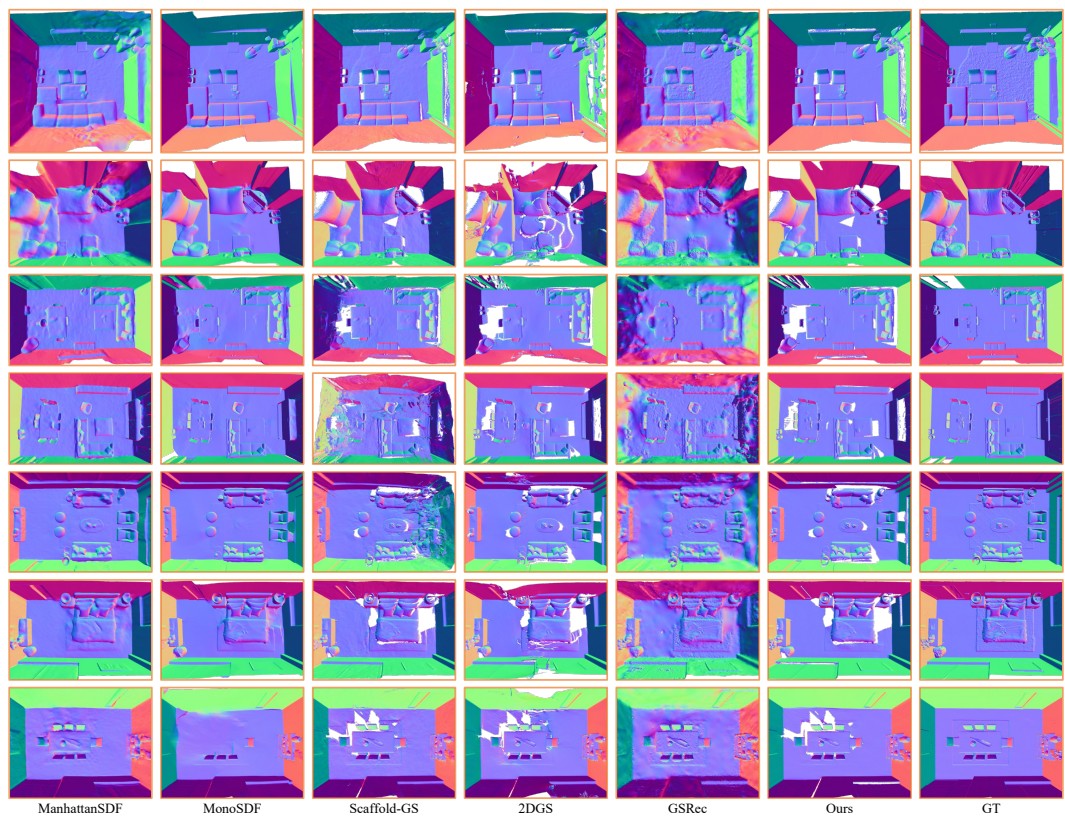

| ManhattanSDF | MonoSDF | Scaffold-GS | 2DGS | GSRec | Ours | GT |

Figure 11: **Qualitative comparison of surface reconstruction on Replica [52].**

