# OpenReview forum: "AtlasGS: Atlanta-world Guided Surface Reconstruction with Implicit Structured Gaussians"
_NeurIPS.cc/2025/Conference — NeurIPS 2025 poster_

### Official Review · Reviewer_p8vj · 2025-06-26

**Clarity:** 3
**Significance:** 2
**Originality:** 2
**Rating:** 4
**Confidence:** 4

**Summary:**

* For the reconstruction of indoor and urban scenes, a geometric prior that can properly restore low-texture regions is necessary, and a new representation is required that overcomes the limitations of GS and SDF.
* First, a good geometric prior can be created by introducing the concept of the Atlanta world model. The Atlanta world model assumes that the vertical direction appears only in the direction perpendicular to the surface, and is applied to most indoor or urban scenes. Using this assumption, proper surface reconstruction is possible even in low-texture regions through regularization.
* The authors proposed an implicit-structured Gaussian representation that can perform efficient rendering while preserving details effectively. By embedding an implicit voxel grid, both the accuracy and efficiency of surface reconstruction are improved.

**Questions:**

In Equation 10, many hyperparameters are used to calculate the loss. I wonder if there is an analysis on how the performance changes depending on the value of each hyperparameter. Also, $\lambda_{sem}$ in L202 should be changed to $\mathcal{L}_{sem}$.

**Ethical Concerns:**

["NO or VERY MINOR ethics concerns only"]

**Final Justification:**

My major concern is the Atlanta world assumption would hinder the performance of other scenes where the ratio of such segments (floor, wall, and ceiling) is not high. During the rebuttal, the authors demonstrated that their model does not affect the performance even if the most of the scene consists of objects where Atlanta world assumption does not hold. Also, the authors acknowledged that the proposed module will cost a computational overhead but showed the overhead is marginal and it still can run in real time.

**Limitations:**

yes

**Quality:**

3

**Strengths And Weaknesses:**

## Strengths
* The authors demonstrated higher reconstruction quality than existing 3D reconstruction models by utilizing the Atlanta world model, which well represents the characteristics of cuboid-shaped indoor rooms or urban cities consisting of buildings.
* The authors stated the limitations of existing 3D representations and proposed a mixed version called implicit-structured Gaussian splitting.

## Weaknesses
* The prime concern of the work is the limited scope of the proposed model.  As stated in the paper, since this method is based on the Atlanta world model assumption, it cannot be applied to more complex scenes with complicated objects. When experimenting with a dataset where the Atlanta assumption is valid, it is trivial to observe that the proposed method outperforms existing methods. However, the scenarios to which this assumption is applied are quite limited, and the performance of AtlasGS may be inferior to existing models in real-world scenarios. The author should demonstrate that the model can achieve comparable performance to more general scenes with complex objects, thereby validating the contribution of the work.
* In implicit structured GS, is a Gaussian predicted for every voxel in a predefined voxel grid? If so, what is the total number of voxels ($N_v$) and the number of GSs estimated for each voxel ($K$)? If a fixed number of GSs is created for every voxel instead of estimating valid anchors (Scaffold-GS), it will generate many redundant Gaussians and become memory-inefficient. This can be seen by comparing the memory cost difference between AtlasGS and other models.
* How are the values of depth and normal calculated? Are they computed through alpha-blending, as in Equation 1? If so, the rendered depth and normal vector might be vulnerable to the attribute of Gaussian splats in the back, which should not be considered in real measurements.

---

> ### Author Rebuttal · Authors · 2025-07-31
>
> We thank Reviewer p8vj for the valuable review comments. We are encouraged that you found our work to have higher reconstruction quality. And we have to apologize for the typos in our paper, and we will fix them in the final version. We address your concerns in detail below.
>
> -----
>
> ### **The limited scope of the proposed model.**
> > W1: The prime concern of the work is the limited scope of the proposed model. As stated in the paper, since this method is based on the Atlanta world model assumption, it cannot be applied to more complex scenes with complicated objects. When experimenting with a dataset where the Atlanta assumption is valid, it is trivial to observe that the proposed method outperforms existing methods. However, the scenarios to which this assumption is applied are quite limited, and the performance of AtlasGS may be inferior to existing models in real-world scenarios. The author should demonstrate that the model can achieve comparable performance to more general scenes with complex objects, thereby validating the contribution of the work.
>
> Our method specifically targets the challenging problem of surface reconstruction in indoor and urban scenes, environments prevalent in daily life and often poorly addressed by existing literature. General-purpose methods, such as 2D Gaussian Splatting (2DGS), frequently falter in these settings due to the common presence of low-texture regions. The integration of an Atlanta-world model provides a crucial global geometric constraint, ensuring consistent surface reconstruction in these difficult scenarios.
>
> However, it's important to clarify that this focus does not restrict our method's applicability. To demonstrate its performance in general scenes, we conducted a comparative analysis on the Tanks and Temples dataset, a standard benchmark for general-purpose reconstruction. Our reported F-score results indicate that our method achieves comparable performance to established techniques, thereby addressing concerns regarding its generalization capabilities.
>
> |    F-score   | 2DGS | Ours |
> |-------------|------|------|
> | Barn        | 0.41 | 0.40 |
> | Caterpillar | 0.23 | 0.20 |
> | Truck       | 0.45 | 0.55 |
> | Couterhouse | 0.16 | 0.14 |
> | Meetingroom | 0.17 | 0.22 |
> | Ignatius    | 0.51 | 0.47 |
> | mean        | 0.32 | 0.33 |
>
> ### **Memory analysis.**
>
> > W2: In implicit structured GS, is a Gaussian predicted for every voxel in a predefined voxel grid? If so, what is the total number of voxels and the number of GSs estimated for each voxel? If a fixed number of GSs is created for every voxel instead of estimating valid anchors (Scaffold-GS), it will generate many redundant Gaussians and become memory-inefficient. This can be seen by comparing the memory cost difference between AtlasGS and other models.
>
> In implicit structured Gaussian, a fixed number $K$ of Gaussians is predicted for each voxel in the voxel grid; however, the voxel grid is not predefined. We initialize the voxel grid with a sparse point cloud, then we generate new voxels with gradient-based densification following Scaffold-GS, and prune the voxel if the opacities of all the Gaussians are lower than a threshold. In our experiments, we set $K$ to 10.
>
> When compared to Scaffold-GS, our method consumes slightly more memory during training, which is a deliberate trade-off. Scaffold-GS reduces memory by estimating only a subset of valid anchors, but this strategy can inadvertently mask visible Gaussians, leading to depth inconsistencies across views and a slight degradation in performance.  But compared with 2DGS, our method is relatively memory-efficient, because 2DGS grows redundant Gaussians in a single voxel, while ours only allows limited Gaussians.
>
> |   Memory   | scene0050_00(Indoor) | matrixcity block6 (Urban) |
> |-------------|---------------------------|-------------------|
> | 2DGS | ~3.6GiB| ~12GiB |
> | GSRec       | ~2GiB / ~20GiB             | ~2.6G /  ~20GiB   |
> | Scaffold-GS | ~1.3GiB                     | ~3.5 GiB             |
> | ours        | ~2GiB                     | ~3.5 GiB            |
>
>
> ### **Biased depth caused by alpha blending.**
> > W3: How are the values of depth and normal calculated?
>
> In our framework, we use mean depth and mean normal, which are obtained from alpha blending as mentioned in 2DGS.
> A well-known artifact in volume rendering, used by both NeRF and Gaussian Splatting, is biased depth. Because the rendered depth for a pixel is a weighted average of all samples along a ray, it can be influenced by points in front of or behind the true surface.
> To alleviate biased depth, our framework combines two techniques. First, we apply the distortion loss from 2DGS to promote a compact opacity distribution. Second, our 3D Global Planar Regularization aligns floor and ceiling Gaussians with a single surface by explicitly regularizing their positions.
>
> ### **Hyperparameter analysis.**
> > Q1: How the performance changes depending on the value of each hyperparameter?
>
> To analyze the impact of key hyperparameters, we conducted a series of ablation studies on the ScanNet (scene0050) dataset. We focused on the weights for the depth prior $\lambda_{depth}$，normal prior  $\lambda_{normal}$, distortion loss $\lambda_{dist}$, and our regularization loss $\lambda_{reg}$. When changing one value, the other one remains the same. The results, presented in the tables below, demonstrate that our chosen hyperparameters are well-justified and that each component contributes positively to the final reconstruction quality.
>
> The distortion loss encourages compact ray sampling along the viewing direction. Increasing its weight from 1.0 to 10.0 results in a noticeable improvement in all metrics. This suggests that a stronger distortion penalty leads to a more coherent and less noisy surface representation. The performance saturates when increasing the weight further to 100.
>
> | scene           | Acc  | Comp | Prec  | Recal | F-score |
> |-----------------|------|------|-------|-------|---------|
> | $\lambda_{dist}$ = 1.0   | 3.40 | 3.68 | 81.15 | 78.73 |   79.92 |
> | $\lambda_{dist}$ = 10  | 3.35 | 3.58 | 82.33 | 79.72 |   81.01 |
> | $\lambda_{dist}$ = 100 | 3.26 | 3.63 | 83.00 | 79.13 |   81.02 |
>
> Our proposed regularization term also plays a vital role. With a minimal weight of 0.001, the performance is suboptimal. As we increase the weight, performance improves, peaking at $\lambda_{reg}$=0.1 with the best F-score of 81.02. However, if the weight becomes too large, the performance begins to decline. This is because an excessively large loss of weight can pull the structural Gaussians too strongly towards the initial planes, potentially causing the entire structure to converge to an incorrect plane.
>
> | scene               | Acc  | Comp | Prec  | Recal | F-score |
> |---------------------|------|------|-------|-------|---------|
> | $\lambda_{reg}$ = 0.001 | 3.59 | 4.12 | 78.90 | 74.80 |   76.80 |
> | $\lambda_{reg}$ = 0.05  | 3.32 | 3.60 | 82.34 | 78.22 |   80.23 |
> | $\lambda_{reg}$ = 0.1   | 3.26 | 3.63 | 83.00 | 79.13 |   81.02 |
> | $\lambda_{reg}$ = 0.2   | 3.41 | 3.83 | 80.72 | 77.39 |   79.02 |
>
> As the following tables show, incorporating and appropriately weighting geometric priors is crucial for high-quality surface reconstruction. These priors improve the results by directly affecting the Gaussian positions and by helping to train the plane indicators, which provide better structural regularization for the scene.
>
> | scene             | Acc  | Comp | Prec  | Recal | F-score |
> |-------------------|------|------|-------|-------|---------|
> | $\lambda_{depth}$ = 0.01 | 3.80 | 4.29 | 77.68 | 73.81 |   75.70 |
> | $\lambda_{depth}$ = 0.1  | 3.40 | 3.96 | 80.01 | 75.69 |   77.79 |
> | $\lambda_{depth}$ = 0.2  | 3.33 | 3.62 | 82.29 | 78.35 |   80.27 |
>
>
> | scene               | Acc  | Comp | Prec  | Recal | F-score |
> |---------------------|------|------|-------|-------|---------|
> | $\lambda_{normal}$ = 0.01  | 3.45 | 3.73 | 79.85 | 76.90 |   78.34 |
> | $\lambda_{normal}$ = 0.1   | 3.26 | 3.63 | 83.00 | 79.13 |   81.02 |
> | $\lambda_{normal}$ = 0.2   | 3.30 | 3.68 | 83.12 | 79.63 |   81.34 |

---

> > ### Comment · Reviewer_p8vj · 2025-08-04
> >
> > First, I would like to sincerely appreciate for the authors' considerable efforts in addressing the rebuttal. The experimental results, particularly those comparing the results by changing the coefficient, have been very helpful in resolving my concerns. Additionally, the clarification regarding how the number of Gaussians is determined has also resolved my doubts (W2).
> >
> > However, I still have some concerns regarding the generalizability of the Atlas world assumption. While the first table shows that AtlasGS demonstrates comparable performance (F-score) in certain scenes, a closer examination of the scene characteristics reveals that when the Atlas world assumption applies well, such as in the case of the Meeting room, performance tends to be strong. Still, in scenes with more complex objects, like Ignatius, the Atlas world assumption appears to have a negative impact. Furthermore, since the F-score metric relies solely on the positions of the Gaussian splat centers, the otehr information, such as the normal vector direction of the GS that could be more heavily influenced by the Atlas assumption, could have larger errors.
> >
> > Additionally, while you have outlined methods to mitigate the errors in depth and normal information, I believe there are some limitations to these approaches. Reducing the number of Gaussians for memory efficiency, for instance, would also diminish the effectiveness of the distortion loss. Similarly, the Global planar regularization method is applicable primarily to scenes where the floor and ceiling can be clearly distinguished, which significantly limits the scope of the model.

---

> > > ### Author Response · Authors · 2025-08-06
> > >
> > > Thank you for your reply. We are glad to hear that we have resolved some of your concerns.
> > >
> > > > However, I still have some concerns regarding the generalizability of the Atlas world assumption. While ...
> > >
> > > First, we didn’t use Atlanta-world assumption and its planar regularization in the evaluation of Tanks and Temples dataset because we want to demonstrate our method on general scenes that don’t necessarily follow Atlanta-world assumption. The results give comparable results to 2DGS. Our structured Gaussian promotes local geometric coherence and provides smoothness of the local Gaussians. This property is advantageous for indoor environments, allowing our method to outperform 2DGS in the case of Meetingroom. However, for scenes with sharper, object-level features like the case of Ignatius, 2DGS achieves slightly better results. As a conclusion, our method not only shows comparable performance to the general method on the general scenes but also overcome the challenging indoor and urban scene reconstruction that the general method fails in.
> > >
> > > > Furthermore, since the F-score metric relies solely on the positions of the Gaussian splat centers, the otehr information, such as the normal vector direction of the GS that could be more heavily influenced by the Atlas assumption, could have larger errors.
> > >
> > > We clarify that our F-score is obtained by comparing the final reconstructed mesh to the ground truth mesh, rather than the centers of Gaussian Splatting. This means F-score also implies the accuracy of rendered normal map since primitives' normal vectors influence the rendered depth, which in turn determines the geometry of the final reconstructed mesh. Furthermore, we explicitly evaluate the accuracy of the reconstructed surface normals relative to the ground truth. And higher is better.
> > >
> > > | Normal-Consistency $\uparrow$ | Ours | 2DGS |
> > > |--------------------|------|------|
> > > | Barn               | 0.88 | 0.89 |
> > > | Caterpillar        | 0.72 | 0.75 |
> > > | Courthouse         | 0.68 | 0.72 |
> > > | Ignatius           | 0.84 | 0.85 |
> > > | Meetingroom        | 0.65 | 0.68 |
> > > | Truck              | 0.84 | 0.84 |
> > > | average            | 0.77 | 0.79 |
> > >
> > > We report the metric of Normal-Consistency on these scenes, and our method is also comparable with 2DGS.  Normal-Consistency is defined in MonoSDF Sec A.4.
> > >
> > > > W3： the rendered depth and normal vector might be vulnerable to the attribute of Gaussian splats in the back, which should not be considered in real measurements.
> > >
> > > > Additionally, while you have outlined methods to mitigate the errors in depth and normal information, I believe there are some limitations to these approaches.I believe there are some limitations to these approaches.
> > > >
> > >
> > > We acknowledge that “errors in depth and normal information”, such as depth bias, are limitations of current methods. However, the issue of biased depth is a long-standing problem in NeRF and Gaussian methods. While biased depth can negatively affect the results, the main issues impacting the quality of surface reconstruction on indoor and urban scenes are the discrete nature introduced by Gaussian primitives and the global inconsistency from monocular geometry priors, which are the primary focus of our work.
> > >
> > > > Reducing the number of Gaussians for memory efficiency, for instance, would also diminish the effectiveness of the distortion loss.
> > >
> > > Sorry, we carefully reviewed the original paper of 2D Gaussian Splatting and didn’t the similar conclusion they have proposed. As mentioned in the rebuttal, our method is more memory-efficient than 2DGS during training. For an indoor scene, we use about 400k Gaussians, which is 60% fewer than 2DGS. GSRec also achieves better results with an even smaller count of ~200k Gaussians. So the reconstruction quality isn't determined by the number of Gaussians.
> > >
> > > > Similarly, the Global planar regularization method is applicable primarily to scenes where the floor and ceiling can be clearly distinguished, which significantly limits the scope of the model.
> > >
> > > Our framework utilizes an off-the-shelf semantic model for element identification, which can sometimes result in inconsistent classifications at segmentation boundaries. To address these misclassifications, we introduce a semantic probability term into our loss function, as detailed in Eq. 6 and Eq. 7. This term penalizes incorrectly classified Gaussian primitives by pushing their associated semantic probability towards zero.
> > >
> > > Moreover, our method achieves performance comparable to general-purpose methods on general scenes like the Tanks and Temples dataset. In contrast, these general methods fail to produce high-quality results for the indoor and urban scenes that are the primary focus of our research. Therefore, our approach offers a key advantage: it excels in our target domains of indoor and urban reconstruction while remaining competitive on general scenes.

---

> > > > ### Comment · Reviewer_p8vj · 2025-08-07
> > > >
> > > > I appreciate your detailed explanations. I acknowledge that I have misunderstood your work in some parts.
> > > > The gist of the paper is to apply the Atlanta world assumption for the part that belongs to floor, ceiling, or wall. The authors train another MLP network to determine which class each Gaussian belongs to.
> > > >
> > > > There are a few questions to confirm that my understanding is right and finalize my remark.
> > > >
> > > > First, if all Gaussian splats are classified as ‘others’ category, is the optimization process identical to original 2DGS? If the number and other properties of Gaussians are the same, is it correct that the same computational overhead (additional runtime) occurs regardless of which class each Gaussian belongs to?
> > > >
> > > > Second, some F-scores and the normal consistency metrics of AtlasGS are slightly worse than original 2DGS. If the model works ideally, the performance of AtlasGS would always be the same or better than 2DGS. What would be the major causes for such differences (wrong classifications of Gaussian splats, regions that the Atlanta world assumption does not hold, or just caused by experimental randomness, etc.)?

---

> > > > > ### Author Response · Authors · 2025-08-08
> > > > >
> > > > > Thanks for your timely reply. We address the questions as follows.
> > > > >
> > > > > > if all Gaussian splats are classified as ‘others’ category, is the optimization process identical to original 2DGS?
> > > > > >
> > > > >
> > > > > No, we have the implicit-structured Gaussian representation compared to 2DGS, which embeds implicit voxel grids within the Gaussian Splatting framework, allowing implicit representation to act as a smooth regularizer while maintaining high-frequency representation benefited from discrete Gaussian primitives.
> > > > >
> > > > > > some F-scores and the normal consistency metrics of AtlasGS are slightly worse than original 2DGS. What would be the major causes for such differences?
> > > > > >
> > > > >
> > > > > We admit there is a slight drop of the F-score in some general scenes, such as Ignatius. We attribute this slight difference to our implicit-structured Gaussian representation with voxel-based densification and pruning. During optimization, our voxel-based representation uses an MLP to ensure that all Gaussians within a single voxel are optimized jointly. This process enforces local geometric coherence and helps model smooth, textureless regions. The trade-off is that this loses some high-frequency details of intricate geometry, such as the case of Ignatius. In contrast, 2DGS optimizes each Gaussian independently. This allows it to better capture high-frequency details on object-level scenes. However, its discrete nature can lead to noisy, protruding artifacts in indoor or urban scenes, as shown in Figure 1.
> > > > > In indoor scenes, the benefit of our approach becomes clear. The quantitative results on ScanNet demonstrate that our method outperforms 2DGS and both of them are under the guidance of geometry priors. The uparrow means higher is better, while the downarrow means lower is better.
> > > > >
> > > > > | ScanNet (w/ geometry priors) | CD$\downarrow$ | F-score$\uparrow$ |
> > > > > | --- | --- | --- |
> > > > > | 2DGS | 12.68 | 39.27 |
> > > > > | Ours (w/o Atlanta Assumption) | 4.10 | 74.23 |
> > > > >
> > > > > > If the number and other properties of Gaussians are the same, is it correct that the same computational overhead (additional runtime) occurs regardless of which class each Gaussian belongs to?
> > > > > >
> > > > >
> > > > > Given the same amount and properties of Gaussians, our method introduces a slight computational overhead compared to 2DGS. This overhead stems from decoding Gaussian attributes from a voxel grid. During each training step, all attributes are decoded from this grid using an MLP regardless of which class each Gaussian belongs to.
> > > > >
> > > > > However, once training is complete, the Gaussians become static. This allows us to pre-extract their view-independent attributes and cache them, which eliminates the decoding overhead during rendering. This pre-extraction significantly reduces the memory footprint and accelerates final rendering performance.
> > > > >
> > > > > The table below compares the rendering performance of our method against other approaches in both indoor and urban scenes. Speeds are reported in frames per second (FPS), showing results with pre-extracted features. We also report the efficiency of our method without pre-extraction (online decoding). And the metrics below are evaluated with CUDA synchronization enabled, without taking into account memory I/O time.
> > > > >
> > > > > | FPS | Scene0050_00 | Block6 |
> > > > > | --- | --- | --- |
> > > > > | GSRec | 264 | 260 |
> > > > > | Scaffold-GS | 282 | 200 |
> > > > > | 2DGS | 267 | 141 |
> > > > > | Ours (Online decoding) | 180 | 90 |
> > > > > | Ours (Per-extracted) | 280 | 191 |
> > > > >
> > > > > The total memory required for inferencing is approximately 1.06 GB for scene0050_00 and 2.24 GB for the matrixcity block6 under the online decoding operation. By pre-extracting the Gaussians, memory usage is reduced by approximately 300 MiB for the indoor scene and 700 MiB for the urban scene.

---

> > > > > > ### Comment · Reviewer_p8vj · 2025-08-08
> > > > > >
> > > > > > Thanks for providing me detailed information.
> > > > > > I think the authors justified that the proposed module gives some advantages in specific scenes, while it requires a marginal overhead in general, which triggers me to raise the score.
> > > > > > The authors are encouraged to clarify the point and show the experiment results on general scenes (as written in the rebuttal) to demonstrate that the model would not give a negative effect on general scenes.

---

### Official Review · Reviewer_QZGu · 2025-06-29

**Clarity:** 3
**Significance:** 2
**Originality:** 2
**Rating:** 4
**Confidence:** 3

**Summary:**

The task of this paper is to reconstruct 3D scenes from multi-view images for indoor and urban scenes. This paper points out two problems in current task: 1) The geometry priors is not global consistent while handling low-texture regions, like planes(floors/ceilings/walls) in the scenes. 2) Gaussian Splatting and implicit SDF filed methods either suffer from discontinuities and inefficient issues. To handle these issue, this paper proposed an Atlanta-world guided implicit-structured Gaussian Splatting which can achieve smoother reconstruction results while preserving high-frequency details and rendering efficiency. Experiment shows that the proposed method well alleviates the problems both qualitatively and quantitatively.

In conclusion, the problem presented in this paper is important for 3D reconstruction, and the proposed scheme is effective. I think it can bring inspiration for the reconstruction community. But due to the weakness and questions comments. I decide to give an border reject for this paper. But I feel open to adjust the score from 3 to higher if the author gives a more clearer response.

**Questions:**

1. I have seen an another work for urban scenes reconstruction[2], they also achieve a great results. What is difference or strength between your work and [2]?
2. Can the latest work VGGT[3] handle the problem of “Traditional multi-view stereo methods struggle on these textureless surfaces due to the lack of distinctive” in Line 22?
3. Was the “plane indicators” mentioned many times the same as  “plane normal”?
4. In Line 253, you mentioned that the scheme need to decode gaussians online? Why need online operation? Isn't gaussians fixed after it has been trained? Further, how about the total GPU memory usage during inference in indoor/urban scenes and how about the additional GPU memory increased by the online decoding operation?
5. Can the proposed scheme used to reconstruct the static scenes in autonomous driving scenario like nuscenes and waymo?
6. In which case the proposed scheme will fail?

**Ethical Concerns:**

["NO or VERY MINOR ethics concerns only"]

**Final Justification:**

All of the questions and weaknesses 2&3, have been adequately addressed and clarified in the rebuttal.
However, weakness 1 and its corresponding rebuttal still leave me confused.
1.I am still confused about the issue of “global inconsistency for monocular geometric priors”. Since Figure 1 does not clarify this issue, it seems that the paper does not formally address it either, neither in a visual nor a mathematical way.
2. In the rebuttal, you mentioned that the problem in Figure 1 “doesn't result from priors but from the discrete nature of Gaussian splatting in low texture regions.” To validate this, you conducted an experiment removing priors from 2DGS, but this experiment does not seem to be related to the issue you are trying to validate. It seems that what you are validating is just whether the performance of 2DGS without priors worsens.
3. Base on the second point, in the rebuttal experiment, you removed priors from 2DGS. Which priors were removed? I also asked in my weakness whether 2DGS uses priors, but it seems that this was not clarified in the rebuttal.

Overall, the rebuttal has addressed most of my concerns, but there are still some issues mentioned above that leave me confused. Therefore, I would like to keep my original rating.

Final justification again.
Actually, after reading the authors' rebuttal, I am still somewhat confused about the geometry prior and related issues. However, considering that I am not a specialist in this particular field (2DGS), other reviewers who are more familiar with this area may have a better understanding of these issues. At the same time, given the paper's strong performance both qualitatively and quantitatively, I believe the methods presented will contribute positively to the field of 2DGS. As a result, I have decided to raise the paper's score from "borderline reject" to "borderline accept."

**Limitations:**

Same with weakness

**Quality:**

2

**Strengths And Weaknesses:**

Strengths*
1. The studied problem is important, the technical details are complete and clearly written which I think can bring insights for the reconstruction community.
2. The proposed method is simple but effective, resulting good in both qualitative visualization results and quantitatively geometry metrics.

Weaknesses*
1. The problem is not clearly presented. According to Line 25 and 27, this paper have mentioned two problems, global inconsistency for monocular geometric priors and failure in not mutually orthogonal structures for Manhattan-world assumption. But according to Figure 1, this paper use 2DGS as baseline to present this problem, but did 2DGS use geometric priors or Manhattan-world assumption? I have searched the 2DGS paper[1], but there are not ‘prior’ or ‘Manhattan’ word in the paper. So Did the problem present by Figure 1 stem at these prior? Actually, I only realize the discontinue issue in ScanNet part in Figure 1.
2. In the experiment, this paper only present the geometry metrics, but how about the semantics metrics after reconstruction like PSNR/SSIM/LPIPS and how about the qualitative results of the semantics reconstruction? Further, can you provide the training time and FPS in Table 3 in urban scenes like Table 2?
3. The rendering efficiency of the scheme seems affected a lot by the newly proposed component.

---

> ### Author Rebuttal · Authors · 2025-07-31
>
> We thank Reviewer QZGu for the valuable review comments. We are encouraged that you found our work both simple and effective, good in both qualitative and quantitative results. We address your concerns in detail below.
>
> ---
>
> ###  **Geometry priors are not the cause of the problem presented by Figure 1.**
> > W1: So Did the problem present by Figure 1 stem at these prior? Actually, I only realize the discontinue issue in ScanNet part in Figure 1.
>
> The broken and distorted surface of 2DGS in Figure 1 doesn't result from priors but from the discrete nature of Gaussian Splatting in low-texture regions. To validate this, we perform an ablation study by removing geometric priors from the 2DGS framework. The results are shown below.
>
> |   | Acc$\downarrow$| Comp$\downarrow$  | Prec$\uparrow$ | Recal$\uparrow$| F-score$\uparrow$|
> |-----------------|-----------|-----------|-------------|-------------|-------------|
> | 2DGS w/ priors  |     11.46 |     13.89 |       43.15 |       36.17 |       39.27 |
> | 2DGS w/o priors |     15.64 |     14.93 |       39.82 |       38.06 |       38.81 |
>
> As shown in the table above, Acc and Comp are lower compared to the case without geometry priors.
>
>
>
> ### **Rendering efficiency and online decoding**
> > Q4: In Line 253, you mentioned that the scheme need to decode gaussians online? Why need online operation? Isn't gaussians fixed after it has been trained? Further, how about the total GPU memory usage during inference in indoor/urban scenes and how about the additional GPU memory increased by the online decoding operation?
>
> > W3: The rendering efficiency of the scheme seems affected a lot by the newly proposed component.
>
> Decoding online means that at each training step, we need to decode all Gaussians from the voxel grid with an MLP. After training, the Gaussians are fixed, and we can preextract the view-independent attributes, which will reduce the memory footprint and improve the rendering performance.
>
> The total memory required for inferencing is approximately 1.06 GB for scene0050_00 and 2.24 GB for the matrixcity block6 with the online decoding operation. By pre-extracting the Gaussians, memory usage is reduced by approximately 300 MiB for the indoor scene and 700 MiB for the urban scene.
>
> While preparing this rebuttal, we re-validated our experiments and discovered a CUDA-synchronization error with considering I/O in our original rendering speed evaluation. We sincerely apologize for this oversight and provide the corrected results below.
>
> The following table compares the rendering performance (in FPS) of our method against other approaches on both indoor and urban scenes. We report speeds using pre-extracted features and speeds in brackets that include online feature decoding.
>
> |    FPS     | Scene0050_00 | Block6 |
> |-------------|---------------------------|-------------------|
> | GSRec       | 264                       | 260               |
> | Scaffold-GS | 282                       | 200               |
> | Ours (Online decoding)       | 180                | 90         |
> | Ours (Per-extracted)       | 280                | 191         |
>
> On both indoor and urban scenes, the online decoding operation leads to a significant performance drop; however, with pre-extraction, the rendering performance is comparable to our methods.
>
>
> ### **Rendering, semantic segmentation quality, and training time, rendering speed for urban scenes.**
> > W2: In the experiment, this paper only present the geometry metrics, but how about the semantics metrics after reconstruction like PSNR/SSIM/LPIPS and how about the qualitative results of the semantics reconstruction? Further, can you provide the training time and FPS in Table 3 in urban scenes like Table 2?
>
> We report the rendering quality and semantic reconstruction quality in Section A.5 of the supplementary material. And here're the results excerpted from the supplementary material.
>
> The following table shows the rendering quality and semantic quality, respectively. And IoUw, IoUf, IoUc represent the intersection of the union of the wall, floor, and ceiling, respectively.
>
> | Replica         | PSNR  | SSIM   | LPIPS  | ScanNet++       | PSNR  | SSIM   | LPIPS  |
> |-----------------|-------|--------|--------|-----------------|-------|--------|--------|
> | ScaffoldGS   | 38.08 | 0.9660 | 0.0961 | ScaffoldGS   | 18.25 | 0.7749 | 0.2764 |
> | 2DGS         | 41.59 | 0.9823 | 0.0464 | 2DGS         | 21.87 | 0.8114 | 0.3060 |
> | DN-Splatter | 29.02 | 0.8967 | 0.2312 | DN-Splatter | 22.76 | 0.8226 | 0.2971 |
> | GSRec       | 36.00 | 0.9574 | 0.1205 | GSRec       | 22.96 | 0.8314 | 0.2708 |
> | Ours            | 39.58 | 0.9756 | 0.0766 | Ours            | 22.51 | 0.8321 | 0.2517 |
>
> | Replica     | IoUw  | IoUf  | IoUc  | ScanNet++   | IoUw         | IoUf         | IoUc         |
> |-------------|-------|-------|-------|-------------|--------------|--------------|--------------|
> | Mask2former | 0.628 | 0.823 | 0.900 | Mask2former | 0.6945714286 | 0.7802857143 | 0.7674285714 |
> | Ours        | 0.701 | 0.846 | 0.927 | Ours        | 0.7317142857 | 0.8581428571 | 0.7771428571 |
>
> We use block6 in MatrixCity to test the performance and training time. The training time is 37 minutes, and the rendering speed could reach 191 FPS with per-extraction.
>
> ### **Differences or strengths between our work and [2].**
> > Q1: What isthe  difference or strength between your work and [2]?
>
> Sorry, there's no reference about [2]. We suppose that you may refer to CityGaussian, which is an urban scene reconstruction method. These urban scene reconstruction methods mainly focus on the problem that is how to reduce memory footprint and maintain the rendering efficiency, and some partitioning or compression methods have been developed. And in our method, we mainly focus on the surface reconstruction and don't take these problems into consideration.
>
> ### **The reconstruction quality of VGGT.**
> > Q2: Can the latest work VGGT[3] handle the problem of “Traditional multi-view stereo methods struggle on these textureless surfaces due to the lack of distinctive” in Line 22?
>
> While VGGT shows powerful reconstruction ability and short inference time, it can provide the desired reconstruction quality on these textureless regions, in which the geometry is simple and easy to estimate. We perform the experiments on ScanNet and Replica with 50 images at a resolution of 518 due to limited GPU memory. Before we evaluate the geometry reconstruction quality, we first use the input pose and ground truth pose to align the point cloud from VGGT, and then use ICP to refine the transformation matrix. The reconstruction quality is shown as follows. As a result, VGGT shows comparable results and higher speed with feed feed-forward model.
> | Replica/ScanNet  | Acc$ \downarrow $| Comp$ \downarrow $  | Prec$ \uparrow $ | Recal$ \uparrow $| F-score$ \uparrow $|
> |-----------------|-----------|-----------|-------------|-------------|-------------|
> | VGGT  | 3.14/3.76 | 8.59/6.14 | 84.99/84.99 | 73.06/73.29 | 78.39/76.40 |
> | Ours  | 2.25/3.62 | 4.08/3.93 | 93.18/80.31 | 82.22/75.85 | 87.35/77.98 |
>
> ### **The “plane indicators” mentioned many times are not the same as “plane normal”.**
> > Q3: Was the “plane indicators” mentioned many times the same as “plane normal”?
>
> Plane indicators are not plane normal but the plane equation. As mentioned in L165, we define explicit plane indicators with a gravity direction and two distance offsets to represent the floor and ceiling.
>
> ### **More results on the autonomous driving scenario.**
> > Q5: Can the proposed scheme used to reconstruct the static scenes in autonomous driving scenario like nuscenes and waymo?
>
> We perform our experiments on 3 static sequences in the Waymo Open Dataset and compare our method with 2DGS, and evaluate the reconstruction results with the GT lidars. We report the Chamfer Distance in meters and F-score with a 0.5m threshold. As shown in the table below, our method shows better accuracy of surface reconstruction than 2DGS.
>
> | Waymo | CD(m)           | F-score     |
> |-------|--------------|--------------|
> | 2DGS  | 1.29 | 44.55 |
> | Ours  | 0.96 | 63.43 |
>
> ### **The proposed scheme will fail when the segmentation model predicts the wrong label.**
> > Q6: In which case the proposed scheme will fail?
>
> Our method uses a segmentation model to distinguish structural elements, such as walls, ceilings. If the segmentation model predicts the wrong label, such as walls being recognized as ceilings, the plane indicators will be biased by wrong-labeled structures, and the surface reconstruction quality of these structural elements may be degraded.

---

> > ### Comment · Reviewer_QZGu · 2025-08-05
> > **Response to the author rebuttal**
> >
> > All of the questions and weaknesses 2&3, have been adequately addressed and clarified in the rebuttal.
> >
> > However, weakness 1 and its corresponding rebuttal still leave me confused.
> > 1.I am still confused about the issue of “global inconsistency for monocular geometric priors”. Since Figure 1 does not clarify this issue, it seems that the paper does not formally address it either, neither in a visual nor a mathematical way.
> > 2. In the rebuttal, you mentioned that the problem in Figure 1 “doesn't result from priors but from the discrete nature of Gaussian splatting in low texture regions.” To validate this, you conducted an experiment removing priors from 2DGS, but this experiment does not seem to be related to the issue you are trying to validate. It seems that what you are validating is just whether the performance of 2DGS without priors worsens.
> > 3. Base on the second point, in the rebuttal experiment, you removed priors from 2DGS. Which priors were removed? I also asked in my weakness whether 2DGS uses priors, but it seems that this was not clarified in the rebuttal.
> >
> > Overall, the rebuttal has addressed most of my concerns, but there are still some issues mentioned above that leave me confused. Therefore, I would like to keep my original rating.

---

> > > ### Author Response · Authors · 2025-08-09
> > >
> > > Dear Reviewer QZGu,
> > >
> > > We hope this message finds you well. As the discussion phase is coming to an end, we would greatly appreciate any further thoughts or clarifications you may be willing to share. If there are concerns that you feel remain unresolved, we would be more than happy to engage and provide additional clarification. Thank you again for your time and valuable feedback.
> > >
> > > Best regards, authors

---

> ### Author Response · Authors · 2025-08-06
>
> I am truly sorry for our oversight. Due to a mistake on our part, you were not included in the "Readers" list, but we selected "Reviewers Submitted" into the "Readers" list, and as a result, our previous response may not be visible to you.
>
> Thanks for your reply, and we apologize for not explaining the issue clearly.
>
> > I am still confused about the issue of “global inconsistency for monocular geometric priors”. Since Figure 1 does not clarify this issue, it seems that the paper does not formally address it either, neither in a visual nor a mathematical way.
> >
>
> Our geometry priors are obtained from monocular estimation models, not from multi-view stereo (MVS). Since each prior is predicted from a single image, the normal vectors for a given surface are frequently inconsistent when transformed into the global coordinate system. This lack of global consistency is a common challenge for methods that rely on monocular geometry.
>
> Figure 1 also demonstrates this issue of global inconsistency with examples from the ScanNet dataset, such as the noisy surfaces on the wall above the sofa (below the yellow box) and the floor beneath the chair. To address this, we adopt an Atlanta world assumption, which allows us to optimize these textureless areas using the Atlanta guided planar regularization, both in 3D and 2D. This technique results in noticeably smoother and flatter surfaces compared to those produced by 2DGS.
>
> > In the rebuttal, you mentioned that the problem in Figure 1 “doesn't result from priors but from the discrete nature of Gaussian splatting in low texture regions.” To validate this, you conducted an experiment removing priors from 2DGS, but this experiment does not seem to be related to the issue you are trying to validate. It seems that what you are validating is just whether the performance of 2DGS without priors worsens.
> >
>
> Our experiments show that regardless of whether geometry priors are used, the surfaces reconstructed by 2DGS visually exhibit similar issues, namely fragmented and discrete surfaces. Since we cannot include figures in the rebuttal, we have only provided quantitative results here. The corresponding visual comparisons will be added in the final version of the paper.
>
> And previous methods, such as MonoSDF, have demonstrated that monocular geometry priors can mitigate the issues associated with textureless regions, despite existing global inconsistencies. However, when it comes to 2DGS with geometry priors, the improvements observed are limited compared to those without geometry priors, which is mainly due to the discrete nature of the approach.
>
> > Base on the second point, in the rebuttal experiment, you removed priors from 2DGS. Which priors were removed? I also asked in my weakness whether 2DGS uses priors, but it seems that this was not clarified in the rebuttal.
> >
>
> Sorry for not explaining in more detail. The prior used in 2DGS is the same as ours, as stated in the Eq. (11) and Eq. (12) in the main paper. We use monocular depth from DepthAnythingv2[1] and monocular normal from StableNormal[2], both for our method and 2DGS.
>
>
> [1] Yang, L., Kang, B., Huang, Z., Zhao, Z., Xu, X., Feng, J., & Zhao, H. (2024). Depth anything v2. *Advances in Neural Information Processing Systems*, *37*, 21875-21911.
>
> [2] Ye, C., Qiu, L., Gu, X., Zuo, Q., Wu, Y., Dong, Z., ... & Han, X. (2024). Stablenormal: Reducing diffusion variance for stable and sharp normal. *ACM Transactions on Graphics (TOG)*, *43*(6), 1-18.

---

### Official Review · Reviewer_pVvf · 2025-06-30

**Clarity:** 3
**Significance:** 3
**Originality:** 3
**Rating:** 5
**Confidence:** 4

**Summary:**

This paper introduces a Gaussian Splatting method for reconstructing a man-made environment with high surface quality, utilizing the Atlanta-world assumption to constrain and guide the scene reconstruction process.

This paper first constructs a sparse feature grid on top of the SfM points, allowing the attributes (also the semantic features) of the Gaussians to be decoded from it. The semantic feature for representing planar structures is learned with supervision from a 2D semantic segmentation model.

Then comes the Atlanta World Guided Planar Regularization, in which the planar region is explicitly indicated with RANSAC, then used for regulating the plane constraint with the proposed loss functions.

**Questions:**

1. I failed to find details about how the predicted 3D mesh is generated from a trained 3D Gaussian Field, whether all Gaussian Splatting-based methods mentioned in the experiments are processed with the same meshing method, or not?

2. The experiments are conducted by evaluating the global geometric reconstruction accuracy. I would like to see the geometric accuracy of non-planar surfaces using AtlantaGS, like running the same evaluation but removing walls, ceiling, and ground floor. From the qualitative result, I see some cases, the constrained planar region also brings better reconstruction quality of other objects nearby the plane. But quantitative results will be even better.

**Ethical Concerns:**

["NO or VERY MINOR ethics concerns only"]

**Final Justification:**

The rebuttal addressed my questions well and helped me to further understand the performance of the proposed method. So I decided to change my rating to accept.

**Limitations:**

Yes.

**Paper Formatting Concerns:**

No major formatting issue noticed.

**Quality:**

3

**Strengths And Weaknesses:**

Strengths:
The methodology of this paper is simple, straightforward, and presented in a clear way. The following experiments on indoor and outdoor datasets showcase the effectiveness of the proposed method. Discussions about the training time and rendering FPS without concealing are also considered a plus mark.

Weaknesses:
Atlanta World assumption is, from my point of view, would break in many cases, e.g., indoor space at the roof floor where the ceiling is not parallel with the ground, a cinema or theater where the ground steps down, a modern design art museum with complex non-plane structured ceiling and even walls which is not plane at all. In that case, the 3D Global Planar Regularization may not stand. A mechanism for dealing with such a situation is not explored in this paper.

---

> ### Author Rebuttal · Authors · 2025-07-31
>
> We thank Reviewer pVvf for the encouraging feedback and insightful review comments. We are glad that you found our work to be simple, straightforward, and of high surface quality. We address your concerns in detail below.
>
> -----
>
> ### **Scenes without Atlanta assumption.**
> >  W1: Atlanta World assumption is, from my point of view, would break in many cases, e.g., indoor space at the roof floor where the ceiling is not parallel with the ground, a cinema or theater where the ground steps down, a modern design art museum with complex non-plane structured ceiling and even walls which is not plane at all. In that case, the 3D Global Planar Regularization may not stand. A mechanism for dealing with such a situation is not explored in this paper.
>
> We agree that the Atlanta-world assumption does not hold in all scenarios, and we acknowledge this as a limitation of our current work. However, our framework includes a mechanism that provides a degree of robustness for scenes that only partially adhere to this assumption.
>
> Our semantic lifting process explicitly includes an "others" category alongside "wall", "floor", and "ceiling". Any surface that is not confidently identified as one of the primary structural elements falls into this "others" category. These Gaussians are subsequently excluded from the planar regularization constraints. This effectively acts as a gating mechanism, ensuring that the strong geometric prior is primarily applied where strong semantic evidence exists.
>
> As for more general scenes without such an assumption, our method still shows comparable results. The following table shows the reconstruction quality on the Tanks and Temples compared with 2DGS.
>
> |  F-score       | 2DGS | Ours |
> |-------------|------|------|
> | Barn        | 0.41 | 0.40 |
> | Caterpillar | 0.23 | 0.20 |
> | Truck       | 0.45 | 0.55 |
> | Couterhouse | 0.16 | 0.14 |
> | Meetingroom | 0.17 | 0.22 |
> | Ignatius    | 0.51 | 0.47 |
> | mean        | 0.32 | 0.33 |
>
>
> ### **Surface extraction methods.**
> > Q1: I failed to find details about how the predicted 3D mesh is generated from a trained 3D Gaussian Field, whether all Gaussian Splatting-based methods mentioned in the experiments are processed with the same meshing method, or not?
>
> All GS-based methods, except for GSRec, use the same meshing method. For GSRec, we follow the meshing method from its original paper. Specifically, GSRec performs Poisson surface reconstruction, and others (including our method) do TSDF-fusion on rendered depth maps using the Open3D package.
>
> ### **Geometric accuracy of non-planar surfaces.**
> > Q2: The experiments are conducted by evaluating the global geometric reconstruction accuracy. I would like to see the geometric accuracy of non-planar surfaces using AtlantaGS, like running the same evaluation but removing walls, ceiling, and ground floor.
>
> We evaluate the accuracy of non-planar surfaces on the ScanNet dataset and use ground-truth semantic masks to exclude planar elements like floors and walls.
>
> The table below provides a quantitative comparison against 2DGS and MonoSDF on ScanNet. Each cell shows the mean/median performance, with arrows indicating the desired direction for each metric ($\downarrow$ lower is better, $\uparrow$ higher is better).
>
>
> |         | Acc $ \downarrow $     | Comp$ \downarrow $      | Prec$ \uparrow $        | Recal$ \uparrow $       | F-score $ \uparrow $     |
> |---------|-----------|-----------|-------------|-------------|-------------|
> | MonoSDF | 7.23/5.23 | 3.76/3.38 | 72.46/75.54 | 78.98/81.56 | 75.22/78.51 |
> | 2DGS    | 9.14/8.00 | 6.41/6.63 | 60.89/64.78 | 62.11/60.78 | 61.37/62.70 |
> | Ours    | **6.95/4.88** | **3.03/2.68** | **74.04/78.26** | **83.88/87.92** | **78.49/82.87** |
>
> As shown above, our method consistently and significantly outperforms both baselines across all five metrics, for both mean and median scores. This underscores our method not only benefits the planar surface in the Atlanta world assumption, but also the non-planar surface.

---

> ### Author Response · Authors · 2025-08-08
>
> Dear Reviewer pVvf,
>
> We hope this message finds you well. As the discussion period is drawing to a close, we would like to kindly check whether there are any remaining questions or suggestions we could further clarify. We're happy to engage further and provide any additional details if needed. Thank you again for taking the time to review our submission and for your thoughtful comments.
>
> Best regards, authors

---

### Official Review · Reviewer_3sjd · 2025-07-07

**Clarity:** 3
**Significance:** 2
**Originality:** 3
**Rating:** 4
**Confidence:** 3

**Summary:**

This paper proposes to address the gaussiona-splat-based surface reconstruction problem with the assumption of the Atlanta world, for achieving better continuity for texture-less area as well as richer high frequency details. This is achieved by incorporating a sparse-volume-grid-based surface representation built on top of the 2DGS representation while maintaining neighbourhood modeling and semantic guidance. Furthermore, to better leverage the Atlanta world assumption, two additional regularization terms are further imposed for failure imposition of the assumption. The experimental results demonstrate overall good results both quantitatively and qualitatively. The paper is well written and executed and clear for reading.

**Questions:**

Please address my questions presented in the "weakness" section above.

**Ethical Concerns:**

["NO or VERY MINOR ethics concerns only"]

**Final Justification:**

I appreciate the authors for taking great efforts addressing my concerns. Overall, I believe the good execution of this work in nature, the adequate evaluation, the idea of particularly addressing scenes with special properties (with no one works on before under the GS context) outweigh the limitation that the methodology proposed mostly for a particular set of scenes. I do believe there are components in this work (e.g. the hybrid representation) could also benefit scenes beyond the Atlanta-world assumption. I believe when the authors mention the monocular priors, they primarily treat it as one viable way to address the issue for textureless regions (not something caused by a baseline method), but it nonetheless brings new problems (not consistent among views). I do not believe it is a critical problem that warrants rejection. I would maintain my score for acceptance.

**Limitations:**

Yes (mostly).

One minor (that might be pressumed to be - so no need for explictly listed) - the proposed approach primarily works for scenes with the Atlanta world assumption.

**Paper Formatting Concerns:**

No issue.

**Quality:**

3

**Strengths And Weaknesses:**

Strength

The paper points out the insufficiency of existing GS-based surface model, in particular like 2DGS with the collapsed ellipsoid onto the surface that leads to discontinuity and lack of details of the surface. These are valuable observations that the authors are motivated to propose their new representation, in particular, a sparse-volume-grid for rich representation power while maintaining budgeted memory footprint, and the MLP “head” for smooth local neighbourhood modeling. While such hybrid feature form is observed also in the NeRF-related literatures, they are nonetheless new in this context.

The authors proposed to impose semantic / attribute predictions that would potentially aid the imposition of the Atlanta world assumption. The proposed regularization terms for maintaining consistent geometry among textureless area are neat, and the overall learning pipeline can lead to promising and high quality reconstruction results.

The authors have conducted with efforts a set of experiments, showcasing that the proposed optimization algorithm can indeed demonstrate faithful local geometric results, especially in those areas where otherwise holes (discontinuity) or irregular / overly smoothed geometric textures presented. Quantitative results on geometric evaluation shows promising outcomes. The pixel reconstruction quality, as expected, occasionally not the best, as the main focus and model design are toward the faithful reconstruction of the geometry. The overall results are convincing.

Weakness

As what the primary goal of this study presented, the proposed approach would be mostly suitable for scenes with the Atlanta world assumptions, meaning the performance might get some level of degradation on datasets otherwise. I am actually pretty curious regarding whether the proposed model could maintain geometric accuracy over scenes without such assumptions, and which of the new components shown in this paper could correspond to the potential improvement on non-Atlanta scenes? (the volume grid?)

While the authors have presented with sufficient arguments regarding their improvement over 2DGS (models that collapse ellipsoid onto the surfaces), the authors nonetheless seemed to provide very brief discussions about to-what-extent that the addressed issues in the context of papers where the Gaussians are regularized to the mesh (e.g. [36]) or an SDF is attached [10]. Are those branches of works can also cope with surface continuity over textureless areas or providing rich geometric textures? Is the volume grid presented in the model that contributes most to the geometric details while the MLP contributed to the continuity (plus semantic guidance)? The relations between what the model has improved and what is proposed would be better elaborated.

---

> ### Author Rebuttal · Authors · 2025-07-31
>
> We thank Reviewer 3sjd for the encouraging feedback and insightful review comments. We are glad that you found our work to be promising and that it has high-quality reconstruction results. We address your concerns in detail below.
>
> -----
>
> ### **Reconstruction quality without such Atlanta assumption.**
>
> > W1: I am actually pretty curious regarding whether the proposed model could maintain geometric accuracy over scenes without such assumptions, and which of the new components shown in this paper could correspond to the potential improvement on non-Atlanta scenes? (the volume grid?)
>
> For scenes without such an assumption, monocular geometry priors cannot provide globally consistent supervision on textureless regions, which may lead to performance degradation as shown in the following table. However, our implicit structured Gaussian will provide local coherent geometry within a voxel and provide better geometry reconstruction quality, which potentially improves the surface reconstruction on non-Atlanta scenes.
>
> |             | CD | F-score |
> |-------------|--------------|------------|
> | 2DGS |  12.68 | 39.27 |
> |  Ours (w/o Atlanta Assumption)       |  4.10| 74.23|
> |  Ours(w/ Atlanta Assumption)        | 3.77 | 77.98 |
>
> For more general scenes without such assumption, we perform the experiments on Tanks and Temples, and Waymo, and compare our implicit structured Gaussian with 2DGS. Our method is still comparable with the previous method as following table. We report the F-score on the Tanks and Temple datasets and the Chamfer Distance on Waymo.
>
> |             | TnT(F-score) | Waymo (CD) |
> |-------------|--------------|------------|
> |  2DGS        | 0.32 | 1.29 |
> |  Ours        | 0.33 | 0.96 |
>
> ### **The relations between what the model has improved and what is proposed.**
>
> > W2: Are those branches of works can also cope with surface continuity over textureless areas or providing rich geometric textures? Is the volume grid presented in the model that contributes most to the geometric details while the MLP contributed to the continuity (plus semantic guidance)? The relations between what the model has improved and what is proposed would be better elaborated.
>
> GSRec and GSDF, two related methods, suffer from issues in textureless regions. GSRec's expensive sampling process, based on sparse Gaussian distributions in these areas, produces noisy surfaces and increases training time and memory. GSDF also has high computational demands, taking about 2 hours and 40 GiB of GPU memory to train its dual Gaussian and NeRF models. Furthermore, its reliance on monocular geometry priors for regularization in textureless regions can lead to globally inconsistent and error-prone results.
>
> In contrast, our implicit-structured Gaussian framework tackles these challenges at the representation level. By using a sparse feature grid and implicit functions to organize Gaussian primitives, we avoid the need for costly external regularizers like SDFs. Our approach is further enhanced by Atlanta world assumption guided planar regularization, ensuring geometric consistency across large, textureless surfaces without the high computational overhead and quality trade-offs of previous methods.
>
> ### **Geometry details and continuity.**
>
> > W2: Is the volume grid presented in the model that contributes most to the geometric details while the MLP contributed to the continuity (plus semantic guidance)?
>
> Our framework utilizes a hybrid approach to accurately model scene geometry.  A sparse feature grid captures the coarse geometric structure, while explicit Gaussians model the fine details. During training, gradients guide the densification and movement of these Gaussians, allowing them to accurately represent the scene's geometry. The MLP decoder provides local continuity. It acts as a continuous function that predicts the attributes of all local Gaussians from the features stored in the grid.
> Because a single, shared MLP generates these attributes for a neighborhood, optimizing one Gaussian inherently influences its neighbors through the shared features and the MLP itself. This ensures the resulting Gaussians are locally coherent and smooth, overcoming the discrete nature of Gaussian splatting.

---

> > ### Comment · Reviewer_3sjd · 2025-08-07
> > **Response to Authors Rebuttal**
> >
> > I appreciate the authors for taking great efforts addressing my concerns. Overall, I believe the good execution of this work in nature, the adequate evaluation, the idea of particularly addressing scenes with special properties (with no one works on before under the GS context) outweigh the limitation that the methodology proposed mostly for a particular set of scenes. I do believe there are components in this work (e.g. the hybrid representation) could also benefit scenes beyond the Atlanta-world assumption. I believe when the authors mention the monocular priors, they primarily treat it as one viable way to address the issue for textureless regions (not something caused by a baseline method), but it nonetheless brings new problems (not consistent among views). I do not believe it is a critical problem that warrants rejection. I would maintain my score for acceptance.

---

### Note · Authors · 2025-08-14

Dear Reviewers, ACs, SACs, and PCs,

We sincerely thank all reviewers for their insightful feedback. We are encouraged that they recognized our key contributions and highlighted several strengths of our work:

- This work provides a neat regularization term for maintaining consistent geometry, and the method is simple but effective. (*Reviewer 3sjd pVvf QzGu*)

- The paper demonstrates superior reconstruction quality over existing 3D reconstruction methods, supported by both quantitative and qualitative results. A comprehensive set of experiments on indoor and outdoor datasets showcases the effectiveness of the proposed method. (*All Reviewers*)

- The paper is well-written, clearly executed, and presents complete technical details, making the work accessible and providing valuable insights to the reconstruction community. (*Reviewers 3sjd, pVvf, QzGu*)

We also briefly clarify the main concerns raised by the reviewers:

- ****Generalizability of our method****: We conduct experiments on the Tanks and Temples and Waymo datasets, and our method provides comparable results on general cases and higher quality reconstructions for both indoor and urban scenes, overcoming a key limitation of general methods that struggle to produce smooth surfaces. As discussed with Reviewer p8vj, our approach offers substantial advantages in these challenging scenarios while introducing only marginal overhead in general cases.

- **Problem presented in teaser**: The problem presented in the teaser is primarily the discrete nature of Gaussians. We conducted an ablation on 2DGS to show that these inconsistent priors do not cause the protruding surfaces. We will provide further illustration and clarification on the problem of global inconsistent geometry priors in our final version.

- **Rendering and Memory Efficiency**: We have provided more details about the memory overhead and rendering performance analysis, which shows our method produces better results with lower memory consumption and provides comparable rendering performance.

Finally, we are grateful to the reviewers for their insightful comments and for acknowledging the effectiveness of our proposed method. We hope that our rebuttal and subsequent discussions have addressed all remaining concerns. In summary, our work offers an efficient solution for high-quality indoor and urban surface reconstruction, and we believe its publication will stimulate further research in this area.

Best regards, Authors

---

### Decision · Program_Chairs · 2025-09-17

**Decision:**

Accept (poster)

**Comment:**

The reviewers value that the "MLP head for regularization is new in this context and neat regularization term" (3sjd), the method is "simple, straightforward, and presented in a clear way" (3sjd), simple and effective (QZGu), important and clearly written (QZGu), that the experients are convincing (3sjd), and prove to be effictive in indoor and outdoor datasets (pVvf).

A major critique from reviewers was the loss of generality due to the Atlanta World assumption (3sjd, pVvf, p8vj). However, in the rebuttal response to reviewer pVvf, the authors have shown that their method also works well on the general Tanks and Temples scenes.

Overall the remainig concerns were resolved and all reviewers vote for accepting the paper. Therefore, the AC recommends the paper for acceptance.